# ROYAL SOCIETY
# OPEN SCIENCE

behaviour/cognition/evolution

joint action, joint commitment, great apes, social grooming, social play, politeness theory

# Evidence of joint commitment in great apes' natural joint actions

Raphaela Heesen[1,2], Klaus Zuberbühler[3,4], Adrian Bangerter[1], Katia Iglesias[5], Federico Rossano[6], Aude Pajot[1], Jean-Pascal Guéry[7] and Emilie Genty[1,3]

[1]Institute of Work and Organizational Psychology, University of Neuchâtel, Switzerland
[2]Department of Psychology, Durham University, UK
[3]Institute of Biology, University of Neuchâtel, Switzerland
[4]School of Psychology and Neuroscience, University of St Andrews, Scotland
[5]School of Health Sciences (HEdS-FR), HES-SO University of Applied Sciences and Arts of Western Switzerland
[6]Department of Cognitive Science, University of California, San Diego, CA, USA
[7]Zoological Park La Vallée des Singes, France

 RH, 0000-0002-8730-1660; AB, 0000-0001-6989-8654;
KI, 0000-0003-1308-1631; FR, 0000-0002-6544-7685

Human joint action seems special, as it is grounded in joint commitment—a sense of mutual obligation participants feel towards each other. Comparative research with humans and non-human great apes has typically investigated joint commitment by experimentally interrupting joint actions to study subjects' resumption strategies. However, such experimental interruptions are human-induced, and thus the question remains of how great apes naturally handle interruptions. Here, we focus on naturally occurring interruptions of joint actions, grooming and play, in bonobos and chimpanzees. Similar to humans, both species frequently resumed interrupted joint actions (and the previous behaviours, like grooming the same body part region or playing the same play type) with their previous partners and at the previous location. Yet, the probability of resumption attempts was unaffected by social bonds or rank. Our data suggest that great apes experience something akin to joint commitment, for which we discuss possible evolutionary origins.

**Author for correspondence:**
Raphaela Heesen
e-mail: heesenr1@gmail.com

## 1. Introduction

When humans interact with each other, they experience a sense of obligation towards their partners, a joint commitment [1,2]. Joint commitment underlies joint actions, from physical acts that require coordinated movements (e.g. moving a piano, dancing together) to communicative acts (e.g. gossiping) [2–5].

Humans monitor each other during interaction and respond to or sanction partners who renege on joint commitments [6].

Empirically, joint commitment manifests itself when ongoing joint actions are interrupted, either due to a partner's inability or unwillingness to continue [7–9] or due to external interruptions by third-party individuals [10,11]. Individuals wishing to interrupt a joint action typically justify the necessity of the interruption, whereas those being interrupted unexpectedly often protest and attempt to re-engage partners [8,9,12–15]. One paradigm to study joint commitment is therefore to have participants engage in a joint action (e.g. a social game) and to then interrupt them or have one partner, a confederate, deliberately interrupt it [7,8,12,16]. Participants' protests or attempts to re-engage their partners to the joint action are interpreted as evidence for joint commitment [8,9,12].

Interruption experiments have mainly been carried out with both humans and non-human great apes in triadic interactions with human experimenters (i.e. involving a shared focus of attention such as an object or device to play with), albeit with contradictory results. In one study, human toddlers (but not chimpanzees) tried to re-engage an experimenter, suggesting that only humans experience joint commitment [8]. In other studies, there was evidence for reengagement attempts by great apes towards passive human partners, suggesting a phylogenetically ancient predisposition [7,17]. These inconsistent findings may be due to methodological differences, ontogenetic differences between studies (due varying developmental stages in which participants were tested), species differences in sociality and social cognition [18,19] as well as the propensities to engage in joint actions with humans. For example, evolved differences in the cooperativeness of humans, chimpanzees and bonobos may be caused by differences in social tolerance [18], proneness for aggression [20], pro-sociality, social attention, emotionality or empathy towards others [21–24], which is thus likely to determine how conspecifics interact socially [25,26]. Regarding ontogeny, human children only start recognizing joint commitment from the age of three [8,15], suggesting that similar ontogenetic patterns may also underlie great ape cognition. Regarding experimental design, the ecological validity of the tasks used in some studies and the use of human experimenters as partners may further explain performance differences across species. Specifically, human-instigated triadic tasks like rolling a ball back and forth [17] or letting a tool drop through a hole with the other partner using a cup to fetch it [8] may be inappropriate for assessing whether apes engage in joint commitment. Purely dyadic interactions (i.e. between two individuals without an object involved) like grooming or chase/rough-and-tumble play might represent more ecologically valid contexts to investigate it.

As a matter of fact, joint commitment is also apparent in human dyadic interactions like conversation [27]. In human joint actions (regardless of whether dyadic or triadic), unilateral interruptions can be perceived as violations of obligations towards partners and, consequently, threats to their face [27,28]. Humans are wary of the social consequences of violating joint commitments and mitigate potential face threats by using communicative strategies that qualify as 'politeness' [28]. Human politeness predominantly involves speech (and to a lesser extent non-verbal communication [29]) and complex forms of self-awareness [30]. In regulating joint commitments, people deal with suspensions and resumptions via utterances such as 'Sorry, I have to deal with this, I'll be right back', or 'Sorry for keeping you waiting' [10,27,31]. The amount of politeness, produced by partners to collaboratively manage each other's face, depends on the duration of the interruption [27] as well as the social distance between the partners and the partners' relative social ranks [28]. For instance, humans tend to be more polite towards partners who are unfamiliar or of higher social rank [28], with politeness typically being expressed via speech acts [32]. Patterns analogue to politeness in humans might also be found in the joint action coordination of great apes (e.g. [33,34]).

For instance, in a recent experiment, we interrupted bonobos engaged in social grooming, either individually (by a keeper calling an individual's name) or as a group (by mimicking an imminent group feeding event). Bonobos communicated with their partners in some cases prior to suspending joint actions [33]—and more so when interacting with a higher ranking or weakly bonded conspecific. These findings suggest that bonobos may experience something akin to joint commitment, which necessitates specific behaviour patterns that, in humans, are interpreted as polite. Bonobos might be aware of the social consequences of unilaterally abandoning partners, suggesting that the ability to engage in some kind of joint commitment may be shared in great apes and humans. We also showed that [34] communication to initiate and terminate joint action in bonobos is similarly affected by social bond (and to much lesser extent rank differences), again in line with the pattern predicted by politeness theory [28], and that this pattern was not evident in chimpanzees. However, questions related to potential species differences with respect to how bonobos and chimpanzees deal with *interruptions* of joint actions still remain open, as well as whether resumption attempts extend to other social activities beyond grooming (e.g. social play, [35]).

In the current study, we investigated the hypothesis that great apes experience joint commitment during naturally occurring joint actions like play and grooming (avoiding human interference).

We developed a study design that circumvented the main confounds and unresolved issues of previous work, i.e. human interference and cross-species interactions, ontogenetic disparities, task artificiality and possible species differences. To this end, we investigated whether joint commitment may underlie joint actions in chimpanzees and bonobos by looking at whether pairs of conspecifics resumed spontaneous, naturally occurring joint actions when they were interrupted by natural causes, rather than artificially instigated ones. Our main interest was in the consequences of such natural interruptions, that is, if, when and how subjects resumed joint actions with their former partners. To this end, we recorded the apes' behaviours during interruptions of their joint actions, caused by either external events (i.e. *External causes*: a sudden noise, a conspecific interrupting, a keeper passing by, food being distributed, or any other environmental or group-related events) or by the partners themselves (i.e. *Internal causes*: partner(s) taking a break from the interaction without other apparent cause for interruption). As it is difficult to distinguish interruptions initiated by the partners themselves (as opposed to those caused by a sudden external event) from a natural ending of the joint action, we focused our analyses on the interruptions that were caused by external events (*External causes*) in our models for testing our predictions.

Should subjects experience a sense of obligation (beyond a mere individual desire to groom or play), we expected them to re-join previous partners, return to the same location (when one or both partners had left the initial location) and continue the previously interrupted behaviour (hereafter called 'continuation of behaviour', defined as either continuing to groom the same body part region or continuing the same play type, like rough-and-tumble or chase play, as before the interruption). The continuation of behaviour implies that subjects recall their own previous behaviours when resuming an interrupted joint action, enabling them to continue where they left off. Indeed, humans typically reconstruct the topic before continuing when interrupted during a conversation [11,27]. Great apes may experience a similar motivation when interrupted in joint actions. By contrast, if great apes had no sense of joint commitment, one might expect them to abandon the interrupted action or continue it with any other conspecific who is close by, especially if they are socially more desirable [36].

We thus predicted that, if subjects experienced a sense of joint commitment with a specific partner, they should not only resume the joint action with the same partner, but also be motivated to reconstruct previous behaviours (e.g. groom the same body part region or engage in the same play type as before the interruption). Furthermore, joint action coordination in humans is governed by social dimensions such as social bond strength and power difference [28]. Although the concept of politeness does not directly extend to non-human animals as such, its evolutionary antecedents may be studied in terms of how social dimensions (friendship and power) affect communication or behavioural processes involved in great ape joint action coordination. If apes' joint action coordination is, similarly to humans', governed by social dimensions, then both rank differences and social bonds should also have measurable effects on the apes' sense of commitment towards the joint action and thus the motivation to resume interrupted joint actions (as well as to continue previous behaviours) with their partner. We thus hypothesize that rates of resumption and continuation of behaviour should be affected by the participants' relationships, i.e. the likelihood of resumption and the continuation of behaviour should be higher for participants that have large rank differences and weak social bonds.

Finally, we predicted species differences in resumption and continuation of behaviour between chimpanzees and bonobos [18,19]. Compared with chimpanzees, bonobos are said to be more perceptive of others' socio-emotional cues [22,23], have increased levels of emotional awareness and empathy [37,38] and act more tolerantly and pro-socially towards others [18,24,39], which suggest that bonobos would also exhibit a higher likelihood of resumption and the continuation of behaviour after interruptions compared with chimpanzees.

# 2. Material and methods

## 2.1. Study species

We collected observational data on two groups of bonobos and three groups of chimpanzees at four zoological parks (see electronic supplementary material, table S1). The bonobo groups ($N = 25$ individuals; range = 4–52 years) were observed at San Diego Zoo ($N = 9$), USA, and at La Vallée des Singes, France ($N = 16$). The chimpanzees ($N = 25$ individuals, range = 5–49 years) were observed at La Vallée des Singes ($N = 7$), La Réserve Africaine de Sigean, France ($N = 9$) and Basel Zoo, Switzerland ($N = 9$).

The subjects at La Vallée des Singes live in large enclosures comprising an outdoor island enclosure with a large forest and climbing structures in grassy areas (8000 m² bonobos, 7500 m² chimpanzees), and

an indoor enclosure with various enrichment and climbing structures (600 m² bonobos, 220 m² chimpanzees). The groups receive food five to six times a day, including daily ratios of primate pellets, fruits and vegetables. Occasionally, both species receive rice, nuts, meat and eggs. Individuals can additionally forage for wild berries and herbaceous vegetation in their outdoor enclosure.

The bonobos at the San Diego Zoo live in an enclosure including both an outdoor (560 m²) and an indoor enclosure (356 m²). The outdoor enclosure consists of numerous climbing structures, a fresh-water stream and climbing enrichment. The group receives diverse food types up to four times a day (primate pellets and cereals, fruits, vegetables and nuts). Occasionally, the group receives additional supplementary foods (honey, peanut butter, popcorn and seeds), often combined with enrichment toys.

The chimpanzees at Basel Zoo live in a building comprising six lodges in the inside enclosure (total 233.3 m²) and two in the outside enclosure (total 477 m²). The lodges include climbing structures, ropes, puzzle boxes and other enrichments, and the subjects can roam between the lodges (two of the lodges are not visible to visitors). The group receives food six times a day, including salad, vegetables, fruits, eggs and primate pellets.

The chimpanzees at La Réserve Africaine de Sigean live in a 200 m² indoor enclosure and a 10 000 m² outdoor island area. During the day, they are kept on the outdoor island that consists of natural vegetation, bushes, trees, sand, a surrounding water pond, climbing structures, bridges and shelters. The group is provided with food six times a day, including vegetables, fruits, primate pellets and nuts, and chicken and eggs once a week.

All groups have access to fresh water at any time.

## 2.2. Data collection

We collected data using focal sampling [40], maintaining a balanced record of focal samples across individuals. We video-recorded focal individuals (15 or 30 min, depending on the site and visibility) using two Panasonic HC-V770 Camcorders with two external Sennheiser unidirectional microphones (MKE-400). Overall, we collected 1298.55 h of observation (bonobos: 600.25 h; chimpanzees: 698.30 h), with on average 30.00 h per bonobo at San Diego Zoo, on average 20.64 h (s.d. = 0.52 h) per bonobo at La Vallée des Singes, 37.01 h (s.d. = 0.45 h) per chimpanzee at La Vallée des Singes, 30.00 h per chimpanzee at Basel Zoo, and 18.08 h (s.d. = 0.42 h) per chimpanzee at La Réserve Africaine de Sigean, (see electronic supplementary material, table S1).

During focal samples, we recorded data on proximity, socially directed approaches and social interactions (grooming and play). We collected proximity data using 5 min scan samples, determining the identities of nearest neighbours (at arm's-length distance) to the focal individual and identities of the other group members present in the vicinity. We continuously recorded socially directed approaches, i.e. whether the focal individual approached another group member or was approached by one, but only if their outcome was positive (resulting in a social interaction or contact sitting) or neutral (resulting in proximate sitting). We also continuously recorded the duration and directionality (who initiated the social interaction) of social grooming and play between the focal individual and a partner. Moreover, we recorded the outcome of agonistic interactions ad libitum, noting the IDs of involved individuals and assigned winner and loser, or recorded whether the interaction ended with an undecided winner. Wins included the following behaviours: aggressing an individual physically or by showing aggressive behaviours, taking away resources, displacing or chasing an opponent. Losses included the following behaviours: fleeing from the opponent, being displaced, showing submissive behaviours such as bared-teeth displays or whimpers, or giving away a resource. In cases where we were unable to define a clear winner or loser (e.g. both showing aggressive behaviours, chasing each other, hitting or slapping another, sharing the resource upon negotiation), we defined outcomes as undecided (tie).

Data on proximity, approaches, social interactions and conflicts were collected live and recorded on an iPad 6 using individually customized templates designed specifically for this study using FileMaker Pro 15 (v. 15.0.3.305).

## 2.3. Social bond and rank difference

To compute the social bond between partners, we used an inverse proxy referring to the social affinity or social bond strength between partners, the dyadic composite sociality index (DSI) [41,42] using the 'socialindices' package in R ([43] https://github.com/gobbios/socialindices). The values of the DSI scale can range from $0 \to \infty$. Lower DSI values indicate weaker bonds, and higher DSI values indicate stronger bonds. Our DSI computation included the measures of partners' duration of grooming and play, their proximity rates (i.e.

having been in arm reach distance) and their approach rates (i.e. focal having approached the partner or the partner having approached the focal). As the San Diego bonobos were not always let out as one coherent group on the same day, we additionally controlled for observation time and co-residency of dyads for this group. The social indices R package ([43] https://github.com/gobbios/socialindices) contains all the functions used to perform this analysis.

The DSI was computed after (2.1) (e.g. [34]), where $d$ includes the numbers/duration of behaviours chosen to select the index (as described above), $f_{ixy}$ indicates the rate/duration of behaviour $i$ for the dyad $xy$ and $\bar{f}_i$ denotes the mean rate/duration of that behaviour $i$ considering all dyads of this sample [41].

$$DSI_{xy} = \frac{\sum_{i=1}^{d} f_{ixy/\bar{f}_i}}{d}. \tag{2.1}$$

To compute rank differences, we computed Elo-ratings [44,45] for individuals of each site using the 'EloRating' package in R [46]. Elo-ratings rely on the outcomes of agonistic interactions between partners and depend on the sequence in which interactions take place over time. Undecided outcomes were counted as a disadvantage to higher ranked individuals. We refer to Elo-ratings as ranks and Elo-rating differences as rank differences [47].

## 2.4. Video coding

We coded joint actions using the computer software ELAN v. 5.2 [48]. We focused on the interruptions of ongoing joint actions of play and grooming, or mixed activities (switching from grooming to play in the same interaction or vice versa). The interruption duration was coded from the moment of suspension until the resumption of the joint action (see electronic supplementary material, figure S3). The moment of suspension corresponded to the moment all movements typical of the activity stopped. For grooming, it corresponded to the cessation of all hand or mouth movements performed to groom the partner's body. For play, suspensions corresponded to the cessation of play-related movements (rough-and-tumble contact play, chasing) and play-related signals (play-faces, laughter). The moment of resumption corresponded to the moment when the movements typical of the activity were resumed. To be considered an interruption, the joint action had to be suspended for more than 3 s, the activity-typical movements had to stop and the attention of one or both partners (measured via gaze direction) had to be distracted away from the joint action, e.g. to be drawn to a different partner, an external event or to a different social or solitary activity. During the interruption, the partners could remain in close proximity (or in contact) or move away from each other. We operationalized the minimum duration of an interruption based on the observation that 3 s is a minimal time window required for an individual to stop all movements characteristic of the activity and produce at least one glance away from the activity. If the attention was not disrupted from the activity but only the movements stopped, we considered the suspension as a 'pause' but not an interruption. Previous studies have used such short time intervals (less than 1 s to 5 s) as a minimal time window of ape response waiting (i.e. the time that elapses before a signaller repeats a signal if they had not received a response by the recipient during communicative exchanges [49–51]). A criterion of 3 s (intermediate between 1 and 5 s) thus represents a reasonable time window to be considered as a perceivable suspension of the actions performed in an ongoing interaction.

### 2.4.1. Resumption (or no resumption) of the interrupted joint action

If the joint action was renewed within 2 min after the interruption, we considered this event as a *resumption of joint action*. If the joint action was renewed after 2 min, but partners had remained in physical contact or exchanged communicative signals in the meantime, we also considered this event as the *resumption of joint action*. If the joint action was not renewed within 2 min, and no communicative signals had been exchanged in the meantime, we considered the previous joint action to be closed (*no resumption of joint action*). The selection of this time window for coding resumptions of joint actions seemed to match the observed distribution of interruption durations (electronic supplementary material, figure S3).

### 2.4.2. Overview of coded variables

We coded the following variables: partners' IDs; duration of joint actions' main body (i.e. the play or grooming activity's duration); the duration of interruption (from onset of suspension of activity-typical movements and attention to the resumption of activity-typical movements); frequency of interruptions within interactions; whether resumptions occurred after interruption (yes/no); whether upon

resumption partners continued the same behaviour they were performing at the moment of suspension (yes/no)—for play, this meant resuming the same play type as before (chase play/rough-and-tumble play), for grooming, this meant continuing to groom the same body part region as before the interruption (see electronic supplementary material, figure S1 for classifications); partner separation, i.e. whether one, both or neither partners moved away by at least a 2 m$^2$ area from the joint action location before the interruption (0: no one moved, 1: one partner moved or 2: both partners moved), and when both did, whether the subjects resumed the joint action at the same location in their enclosure as before the interruption (defined as the 2 m$^2$ area around the original location).

We also distinguished between internal and external causes of interruptions. External causes are sudden events occurring in the environment (e.g. noise of plane or other animals in the zoo) or in the holding area (e.g. noise of holding trap-doors, animals transferred to another enclosure), or with group members (e.g. conspecific interrupting, group travelling, group vocalizing, conflicts), food (arrival or distribution), or keepers and visitors passing by or interacting with the apes. Internal causes are initiated by one (or both) of the interacting partners who simply took a break from the interaction, without any apparent external source of disruption. As it is uncertain whether interruptions following internal causes (partner breaks) can be considered as interruptions *per se* or represent natural endings of the joint action, we focused our analyses (models 1 and 2) on externally caused interruptions (see the summary of internally and externally caused interruptions in table 1, results).

In total, we coded 1180 joint actions, including 329 of social play (bonobos: 169; chimpanzees: 160), 810 of social grooming (bonobos: 239; chimpanzees: 571) and 41 of mixed joint actions (i.e. where partners switched between play, grooming and/or sex; bonobos: 9; chimpanzees: 32). Of the 1180 joint actions, 845 included 2704 interruptions (see electronic supplementary material, table S4 for a detailed summary of interruptions across activity types and species) and 335 did not include any interruptions. Overall, 156 distinct dyads (bonobos: 80; chimpanzees: 76) experienced interruptions while interacting, with on average 17.3 interruptions occurring per dyad ($N = 2704$, s.d. = 15.1, range = 1–74). This yielded 14.5 interruptions on average in bonobo dyads ($N = 1159$, s.d. = 9.7, range = 1–48) and 20.3 interruptions on average in chimpanzee dyads ($N = 1545$, s.d. = 18.7, range = 1–74).

We assessed inter-observer agreement about whether (or not) the same partners resumed the same behaviour and whether (or not) the interaction had been resumed at the same location (2 m$^2$ area) between R.H. and another coder at PhD level, who was blind to the hypotheses and untrained. The test revealed substantial agreement for both the continuation of the same behaviour after interruption ($N = 64$ interruptions, Cohen's $\kappa = 0.71$) and for the resumption at the same location ($N = 67$ interruptions, Cohen's $\kappa = 0.78$).

## 2.5. Computation of descriptive statistics

### 2.5.1. Frequency and duration of interruptions across activity types and species

To assess the frequency of interruptions, we computed the mean rate of interruptions per minute of interaction across activity types. For the computation of rates on joint action resumption, the continuation of behaviour and returning to the original location after interruption, see electronic supplementary material, text S1. We also computed the mean, median and standard deviation (s.d.) of the duration of interruptions after which joint actions were resumed (electronic supplementary material, table S4).

### 2.5.2. Partner separation (after interruptions)

Partner separation indicates whether partners moved away from each other after the interruption occurred (0 = no; 1 = one partner moves farther away than the 2 m$^2$ area from original location and other partner stays; 2 = both move farther away than the 2 m$^2$ area from original location). Proportions of partner separation were only computed for those interruption events that had been resumed, as otherwise the partners eventually left in any case. We excluded out-of-sight events where the separation distance of partners could not be determined ($N = 8$).

## 2.6. Bayesian mixed regression models

We used Bayesian mixed models to test our predictions [52]. We characterized uncertainty by two-sided credible intervals (including lower and upper 95% CrI), referring to the range of probable values.

**Table 1.** Causes of interruptions during natural joint actions (play, grooming and mixed joint actions).

| interruption cause | examples | nature | N | % interruption | % separation[a] | % resumption |
|---|---|---|---|---|---|---|
| partner break[b] | one or both partners stopping the interaction without obvious reason | internal | 1649 | 61.0 | 9.4 | 100.0 |
| conspecific interrupting | a group member interrupting an ongoing interaction by jumping over, displacing, or communicating with one of the partners | external | 387 | 14.4 | 15.9 | 73.1 |
| unspecified | events clearly distracting partners' attention but could not be specified | external | 366 | 13.5 | 11.7 | 91.0 |
| food | food being visible or distributed | external | 95 | 3.5 | 37.0 | 28.4 |
| environment | movements of other animals in enclosures, noises unrelated to holding routines like aeroplanes or other animals in the zoo | external | 71 | 2.6 | 8.3 | 85.9 |
| group vocalizations | whole group or group members vocalizing | external | 53 | 2.0 | 22.7 | 84.9 |
| conflict | conflict between group members | external | 24 | 0.9 | 20.0 | 83.3 |
| visitors | visitors interacting with apes or behaving in ways that attract apes' attention | external | 21 | 0.8 | 5.9 | 81.0 |
| keepers | keeper walking by or interacting with apes | external | 18 | 0.7 | 0.0 | 38.9 |
| holding area | noises from inside the holding building, animal transfers between enclosures, sound of trap-doors or keeper routines | external | 13 | 0.5 | 50.0 | 61.5 |
| group travel | group traveling to another part of the enclosure | external | 4 | 0.2 | 100.0 | 50.0 |
| out of sight[b] | one or both partners not being visible at the start of the interruption | — | 3 | 0.1 | — | — |

[a]One or both partners moved further away than the 2 m$^2$ area of location before interruption, see Material and methods.
[b]Excluded from Bayesian models for hypothesis testing.

Thereby, evidence for an effect in a certain direction (positive or negative) was provided when posterior distributions shifted away substantially from zero in the corresponding direction, as opposed to centring on zero (i.e. corresponding to the null expectation of posterior distributions). Hence, for inference, we calculated 95% credible intervals from the posterior distributions and checked whether 0 was contained in this interval. We also indicated the estimated mean of the posterior distribution (parameter estimate $b$, [52]), denoting the association between the outcome variables and model predictors. For instance, take the main fixed effect of 'species' (model 1, details below). A positive value for $b$ would indicate that chimpanzees—compared with bonobos (as reference level)—exhibit a *higher* probability to resume joint actions. A negative value for $b$ would indicate that chimpanzees—compared with bonobos—exhibit a *lower* probability to resume joint actions. Finally, we also indicated the s.d. of the posterior distributions [52]. For our models, we only considered the 'External cause' data ($N = 1052$), of which we excluded further cases (details below).

### 2.6.1. Model specificities

To test our research questions, we fitted Bayesian generalized mixed models (models 1 and 2) using the Stan computational framework (http://mc-stan.org/), accessed through the *brms* package [52] in R v. 3.5.0 [53]. Each model included four Markov chain Monte Carlo (MCMC) simulations, with 10 000 iterations per chain, of which we specified 2000 iterations as warm-up to ensure sampling calibration. The model diagnostics revealed that the posterior distributions reflect the distribution of the original response values appropriately, R-hat statistics were less than 1.05, the number of effective samples was greater than 100 and the MCMC chains had no divergent transitions (electronic supplementary material, table S2 and figure S2). For all models, we used the default priors of the *brms* package, which were weakly informative with a student's $t$-distribution of three degrees of freedom and a scale parameter of 10 (electronic supplementary material, table S2).

Due to the large age spectrum of individuals within the sample (i.e. range = 4–52 years, mean = 20 years, s.d. = 13 years) and the potential effects of sex, we investigated whether the model fit improved by including age and sex differences as control variables. We ran leave-one-out information criterion (LOOIC) comparisons [54] between the full model (including variables age or sex difference) and the reduced model (excluding variables age or sex difference) and found no substantial differences in the expected log predictive density (electronic supplementary material, table S3). This indicated that the full models did not improve accuracy, which led us to exclude age and sex differences from the final models.

### 2.6.2. Likelihood of joint action resumption and continuation of behaviour by species and activity type

To test for the likelihood of resuming a joint action with the same partner, we fit a first model (model 1), with the dependent variable being 'resumption' or 'no resumption' (fitting a Bernoulli distribution with binary outcome). The fixed effects were interaction terms between the activity type (grooming versus play) and species (bonobos versus chimpanzee), and interaction terms between species and DSI (i.e. as a measure of social bond), and species and rank difference (i.e. as a measure of power difference). We z-transformed both DSI and rank difference for all individuals (both species) to mean = 0 and s.d. = 1. Additionally, we fit random intercepts for interaction ID, dyad ID and external interruption causes. For this dataset, we excluded events that were out of sight ($N = 3$), and only considered interruptions occurring during grooming or play, caused by external causes ($N = 1052$)—i.e. we excluded internal perturbations that were initiated by subjects themselves (partner break). Of these, we further excluded $N = 40$ external interruptions occurring during mixed activity types due to limited sample size (i.e. less than or equal to 10 interruptions in bonobos; see electronic supplementary material, table S4). Thus, model 1 involved $N = 1012$ data points.

### 2.6.3 Likelihood of joint action resumption and continuation of behaviour depending on partners' social bonds or rank differences

Model 2 was fitted to estimate the likelihood of behavioural continuity. We used the same fixed and random effects as for model 1, but the dependent variable was 'continuation of behaviour' or 'no continuation of behaviour' (fitting a Bernoulli distribution with binary outcome). Once again, like for model 1, here we excluded events that were out of sight ($N = 3$), and only considered interruptions occurring during grooming or play, caused by external causes ($N = 1052$). Of these, we again excluded any further external interruptions that occurred in mixed contexts ($N = 40$) due to small sample size in

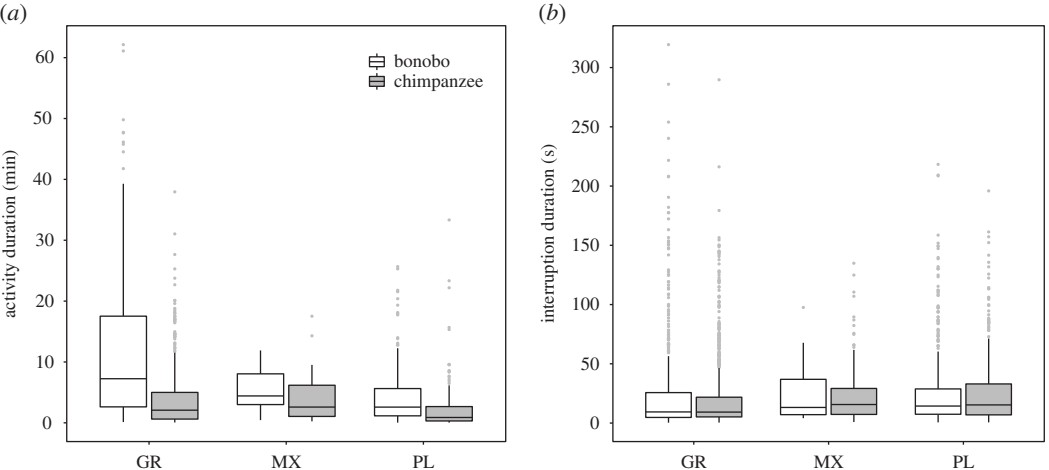

**Figure 1.** Summary statistics for the duration of joint activities (*a*) and interruptions ((*b*) including internal and external causes) across activity types grooming (GR), mixed (MX), play (PL) and species. The lower and upper hinges correspond to the 25th and 75th quartiles, and the middle line to the median. The upper whiskers extend from the hinge to the largest value at most 1.5 × the distance between the first and third quartile. The lower whisker extends from the hinge to the smallest value at most 1.5 × the distance between the first and third quartile. Outlier points beyond the end of the whiskers are plotted individually.

bonobos and interruptions that were not resumed (i.e. we could only determine the likelihood by which partners would continue behaviours when they resumed joint actions; $N = 239$), as well as those that were of insufficient visibility for determining whether the same body part regions were groomed/same play type was played upon resumption ($N = 45$). This yielded a dataset for model 2 with $N = 728$ data points.

# 3. Results

Overall, $N = 845$ of 1180 joint actions contained interruptions (71.6%), which occurred at a rate of 0.74 interruptions per min (s.d. = 1.2, electronic supplementary material, table S5). In total, we recorded $N = 2704$ interruptions (electronic supplementary material, table S4), with some joint actions containing multiple interruptions (mean = 2.3 interruptions per joint action; s.d. = 3.0; range = 1–33). We identified 11 natural causes of interruption (table 1). The most common one was the internal cause of *partner break* ($N = 1649$; 61.0%), followed by external causes ($N = 1052$; 38.9%), of which 65.2% were specified ($N = 686$) and 34.8% were unspecified ($N = 366$; evidenced by attention towards an unspecifiable disturbance). In $N = 3$ further cases, one or both partners were entirely out of sight. Examples of resumptions after various causes can be found in the electronic supplementary material, movies S1–S8 (see [55]).

The mean duration of interruptions (internal and external causes) after which joint actions were resumed was 25.3 s ($N = 2455$, s.d. = 33.6 s, first quartile: 6.4 s; third quartile: 28.3 s; range = 3.0 s–319.4 s), (see electronic supplementary material, figure S3). The mean durations of grooming/play/mixed joint actions, as well as of interruptions (internal and external causes) after which these joint actions were resumed can be found in electronic supplementary material, table S4 and figure 1.

## 3.1. Likelihood of joint action resumption and continuation of behaviour by species and activity type

Across activity types, we found that the mean percentage by which apes resumed joint actions with their original partner was 71.0% ($N = 1052$, s.d. = 26.0%, electronic supplementary material, table S6). The mean percentage by which apes resumed joint actions at the same location after having left their previous 2 m² area was 63.8% ($N = 114$, s.d. = 42.0%, electronic supplementary material, table S7B). Model 1 revealed that the likelihood of resumption was similar across grooming and play (figure 2*a*; electronic supplementary material, table S2; $b = 0.3$, s.d. = 0.3, 95% CrI [−0.3, 0.8]). Although bonobos generally had a higher likelihood of resumption than chimpanzees, the effect was rather weak (figure 2*a*; electronic supplementary material, table S2; $b = −0.5$, s.d. = 0.3, 95% CrI [−1.1,

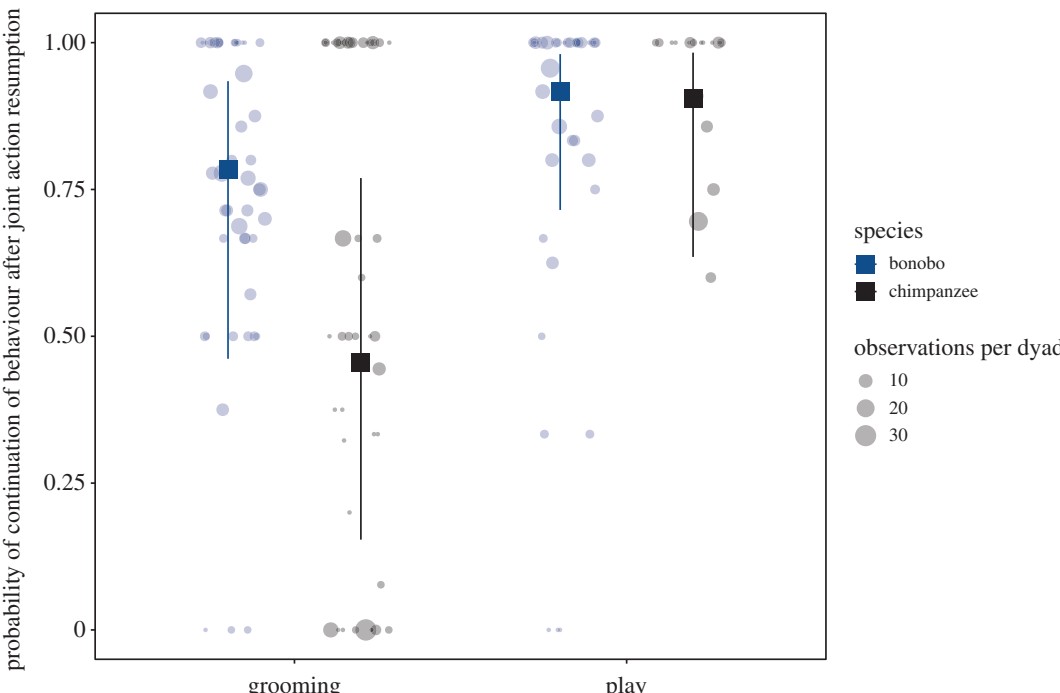

**Figure 2.** Variation of joint action resumption ((*a*) model 1) and the continuation of behaviour ((*b*) model 2) in grooming and playing of chimpanzees and bonobos after external interruptions. Data plots depict the predicted probability of outcome variables for the marginal effects of the interaction term 'joint action × species' for the complete Bayesian generalized linear mixed models. Mixed activities were excluded due to the small sample size. The upper and lower vertical lines correspond to the upper and lower 95% credible intervals, respectively, and the squares represent the posterior means. Each circle corresponds to the proportion of resumption (*a*) or continuation of behaviour (*b*) per dyad, with the size of the circles corresponding to the number of interruptions (*a*) or resumed joint actions (*b*) observed per dyad across activity types.

0.0]); indeed, as visible from figure 2*a*, both species were highly likely to resume joint actions, regardless of the activity type (electronic supplementary material, table S2; $b = -0.0$, s.d. $= 0.5$, 95%CrI [−1.0, 0.9]).

Furthermore, across activity types, we found that the mean proportion of the continuation of behaviours after joint action resumption across dyads was 73.2% ($N = 758$, s.d. = 31.0% and electronic supplementary material, table S7A). However, model 2 revealed that the likelihood of the continuation of behaviours was greater for play than for grooming (figure 2b; electronic supplementary material, table S2; $b = 1.1$, s.d. = 0.4, 95% CrI [0.3; 2.0]). Both species generally tended to continue behaviours more in play than in grooming (figure 2b; electronic supplementary material, table S2; $b = 1.3$, s.d. = 0.8, 95% CrI [−0.1, 2.9]), but chimpanzees were estimated to be less likely than bonobos to continue behaviours overall, most notably during grooming (electronic supplementary material, table S2; figure 2b; $b = −1.4$, s.d. = 0.4, 95% CrI [−2.3, −0.6]).

## 3.2. Likelihood of joint action resumption and continuation of behaviour by social bonds and rank differences

We considered whether the social relationships between partners affected the likelihood of resuming (and continuing behaviours) in naturally interrupted joint actions after external causes. Model 1 revealed no effects of social bonds (electronic supplementary material, table S2; $b = 0.1$, s.d. = 0.2, 95% CrI [−0.2, 0.4]) or rank differences (electronic supplementary material, table S2; $b = 0.0$, s.d. = 0.2, 95% CrI [−0.3, 0.4]) on resumption. Similarly, model 2 revealed no effects of social bonds (electronic supplementary material, table S2; $b = 0.2$, s.d. = 0.2, 95% CrI [−0.2, 0.7]) or rank differences (electronic supplementary material, table S2; $b = −0.1$, s.d. = 0.3, 95% CrI [−0.6, 0.4] on the continuation of behaviour. We also found no evidence for interaction effects between species and these two social variables (electronic supplementary material, table S2).

# 4. Discussion

Empirically, the resumption of an interrupted joint action has been taken as evidence for an underlying sense of joint commitment (e.g. [8,9]), an allegedly uniquely human capacity resulting from an unmatched cooperative nature and motivation to share goals [56]. In the present study, we challenge this view by showing that, following natural interruptions of joint actions, both chimpanzees and bonobos resume joint actions with previous partners, similarly to humans [4,10,11,27]. This was regardless of joint action type and usually consisted of continuing to groom the same body part region or to play the same play type—notably at the same location, indicating some continuity of the suspended joint action rather than the start of a new one. Although we only found weak support for species differences with regard to the overall resumption likelihood (bonobos being only slightly more likely to resume joint actions compared with chimpanzees), species differences were more evident in the *way* by which subjects resumed joint actions (bonobos were more likely than chimpanzees to continue the same behaviour like grooming the same body part region).

Contrary to our predictions, however, the likelihood of resumption and continuation of behaviour were independent of social dimensions (bond strength or rank differences) in both chimpanzees and bonobos. This finding is at odds with recent findings in which bonobos' (yet not chimpanzees') communication efforts to initiate, resume and terminate joint actions were affected by social bonds and—to a lesser extent—rank differences [33,34]. These findings suggested that, at least in bonobos, the communication efforts deployed to coordinate joint action might follow some of the patterns in human interaction, as predicted by politeness theory [28]. In the present study, the fact that the likelihood of resumption and the continuation of behaviour were not affected by social dimensions suggests that the relationship quality between partners does not determine *whether* great apes resume joint actions after interruptions but, more subtly rather, *how* they handle interruptions and resumptions, via communicative acts [33,34]. Future studies should focus on the specific signalling behaviour, produced during suspension and resumption of joint actions, and whether it is relative to relationship quality. At this point, it seems reasonable to conclude that resumption attempts are the default reaction to interruptions of joint action in great apes and humans (see also [33]). To decide whether politeness theory has any heuristic value for primate social cognition, further research into the way the different species communicate about suspension and resumption [33] is needed.

Contrary to our predictions, we also found only weak support for a species difference in the likelihood to resume interrupted joint actions. The fact that both species resumed joint actions at comparably high rates, and regardless of relative rank, social bond and activity type, shows that the resumption of a joint action is a priority in both species. We did, however, find species effects in the

likelihood of continuing the same behaviours as before when resuming joint action (which, in humans, is a crucial aspect of maintaining the integrity of joint actions [10,27]): although there were no marked species differences for social play, bonobos were more likely than chimpanzees to continue grooming their partner's same body part regions as before interruptions, suggesting that—although both species are similarly motivated to resume—they differ in the way by which they resume. The continuation of previous behaviours thus appears to build the basis for resuming joint actions in apes. Indeed, also in humans, reconstructing a topic is a common way of reinstating conversation, with common questions asked by listeners, such as 'You were saying?', or by speakers, such as 'Sorry […], where was I?[…]' [4,11,27].

Nonetheless, it is worth noting that the likelihood of the continuation of behaviour was not the same across activity types, insofar as it was lower in grooming (i.e. grooming the same body part region as before the interruption) compared with social play (i.e. playing the same play type as before the interruption). This difference could be explained by the fact that the range of behaviours that we considered for play continuation (chase play versus rough-and-tumble play) was limited compared with those considered for grooming continuation (10 different body part regions). Another potential explanation might be related to a differential motivation inherent to the type of activity, yet further research is needed to answer this question.

Relatedly, it remains unclear whether apes profit from individual (versus collective) psychological rewards. In humans, joint success appears to outweigh individual benefits, evidenced by findings showing that toddlers help collaborative partners even after they had already received their own reward [57]. However, there is now evidence that also apes (at least bonobos) may profit from such social rewards when engaging in joint action. This is evidenced by a higher likelihood to resume social interactions (social grooming) compared with solitary actions like self-grooming [33]. Whether social play, like social grooming, also elicits higher rates of resumption compared with non-social activities would require further investigation.

Another open question is whether joint commitments are qualitatively different in dyadic versus triadic joint actions. On the one hand, triadic (as compared with dyadic) joint actions might be more complex to process as they require mutual attention to the same object [17]; yet on the other hand, one might argue that triadic joint actions may be simpler to coordinate because of the presence of an object. Indeed, in human joint action, objects can facilitate coordination by enabling a shared perspective [58]. Whether they also facilitate joint commitments in human or non-human joint actions remains an exciting question for future research.

Our data alone cannot be used to ascertain the psychological mechanisms involved in the behaviours at hand. One important remaining question is whether, when resuming joint actions with others, apes (like humans) expect their partners' intentions to align with theirs (i.e. sharing intentions). Recent research showed that apes (like human children) have the basic prerequisites to create bonds with others through shared experiences [59], but differ in the socio-cognitive processes involved in this process [60]. Although apes might be capable of inferring that their goal overlaps with their partner's [61], humans supposedly mutually *know* (or make each other know, as by declarative pointing [61]) that they are sharing their intentions towards the same goal [62]. But how then can we explain that apes resume joint actions? Some promising lines of evidence suggest that the nature of apes' joint commitment is more than just a pursuit of personal goals (where individuals just happen to groom each other because their personal goals overlap). Indeed, ape commitments seem to be based on substantial other-awareness and partner-sensitivity. For instance, we know now that (i) after interruptions, individuals resume the joint action not just with anyone that is in close proximity, but with their original interaction partner (current study and [33]), (ii) individuals do not just resume joint actions in a random way, but they take up previous interaction roles [33], locations and behaviours (e.g. grooming the same body parts, as in the current study), (iii) joint actions are more likely to be resumed than individual actions [33], and (iv) bonobos' communication is dependent on their own responsibility in suspending the joint action, as well as their partner's rank and the social bond they share [33]. Furthermore, chimpanzees and bonobos do not just unilaterally abandon their interaction partners when joint actions end; rather, they go through an interactive process of entering and exiting from joint actions, evidenced by mutual gaze and communicative signal exchanges [34]. Although more evidence will be needed to assess the underlying cognitive abilities involved in those visible patterns of resumptions and sequencing of actions, these new findings show that some aspects of the structure of joint action coordination appear to be shared among *Homo* and *Pan* [34].

One alternative explanation regarding resumption rates should be considered here; it could be argued that what we observed here are not resumptions of interrupted joint actions, but rather mere endings of

joint actions, followed by initiations of new ones (for a similar discussion, see [33]). To circumvent this uncertainty, our predictions were only tested on interruptions caused by external events (e.g. a startling noise in the environment, a conspecific interrupting, a keeper passing by), in which neither partner was responsible for, nor had decided or (presumably) mutually agreed to end the joint action. Furthermore, the fact that bonobos frequently continued the same behaviour as before the interruption (grooming the same body part region/playing the same play type), and at the same location, even after having moved away from the original location, suggests that we are observing resumptions of interrupted joint actions rather than initiations of new ones. Another line of evidence to support this idea is that, after an interruption, bonobos adjust their communication in relation to their previous joint action roles as well as their own responsibility in suspending the activity [33].

A last challenge is to fit our findings into evolutionary theories on joint commitment. Some prominent theories suggest that joint commitment has evolved in the context of obligate collaborative foraging in early *Homo* [63,64]. However, the present study and others [33,34] seem to indicate that both bonobos and chimpanzees may experience some form of joint commitment, suggesting that joint commitment could have evolved earlier than in the genus *Homo* and may have been already present in our common ancestor with *Pan*. But what could be the selective factors that favoured the evolution of joint commitment in bonobos and chimpanzees? Although bonobos are assumed to be more socially tolerant [18] and chimpanzees engage in larger-scale cooperative activities in the wild, such as border-patrolling [65,66], both species regularly engage in complex cooperative activities, notably collaborative hunting [67–70]. Moreover, the establishment of female alliances in bonobos permits collective retaliation against male aggression, and bonobo males do engage in coalitionary attacks against out-group males [71,72]. Further interdisciplinary research should be conducted to investigate environmental selective pressures that might have led to the evolution of joint commitment in these species. Moreover, research should focus on studying joint commitment in other ape species like gorillas and orangutans, as well as other primates more distantly related to humans (e.g. species that engage in cooperative breeding [73]) or even other social animal species outside the primate order.

# 5. Conclusion

In conclusion, our findings suggest that great apes might have some sense of joint commitment when engaging in joint actions with conspecifics. Both bonobos and chimpanzees frequently resumed joint actions with their initial partners and continued previous behaviours after interruptions. This ability to experience or engage in joint commitments—as shared mental state or as visible interaction process [34]—may thus be part of ancestral primate roots. Whether the underlying mental capacities align with those of humans or not, cannot be decided with the current data. Our impression is that much is still to be studied and learnt from the natural joint actions of great apes about how their social minds operate.

Ethics. We received ethical agreement for this study from the Commission d'Ethique de la Recherche of the University of Neuchâtel (agreement no. 01-FS-2017), the internal ethical committee of La Vallée des Singes, San Diego Zoo Global Institutional Animal Care and Use Committee (Project no. 17-007), the internal ethical committee of the Réserve Africaine de Sigean, and the Kantonales Veterinäramt BS at Basel Zoo.

Data accessibility. Datasets and R codes, as well as example movies (S1–S8), have been uploaded to an online repository (http://doi.org/10.6084/m9.figshare.14891865) [55].

Authors' contributions. R.H. designed the study, collected and analysed the data, coded data of bonobos and wrote the first draft of the manuscript; A.B., F.R. and K.Z. designed the study and edited the manuscript; K.I. analysed the data and provided statistical advice; J.-P.G. provided logistic support at La Vallée des Singes; A.P. coded the data of chimpanzees; E.G. designed and coordinated the study, coded and checked the coding and edited the manuscript.

Competing interests. The authors declare no competing interests.

Funding. The present research was supported by the Swiss National Science Foundation (grant no. CR31I3_166331 awarded to A.B. and K.Z.).

Acknowledgements. We thank Adrian Baumeyer at Basel Zoo, Dean Gibson, Mike Bates and Kim Livingstone at San Diego Zoo, Emmanuel Le Grelle and the Conservatoire pour la Conservation des Primates at La Vallée des Singes, Antoine Joris and Marielle Vandenbunder-Beltrame at La Réserve Africaine de Sigean, and all the dedicated animal keepers, for letting us conduct our observations at their sites, and for providing invaluable support during the study periods. We are also grateful for the help of our trained research assistants Cindy Maslarova and Laura Perrenoud. Additionally, we thank Martin Götz for discussions and input regarding Bayesian statistics, and Quentin Gallot for his assistance with the inter-rater agreement coding.

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
