## [Peer Review File · Royal Society Open Science]

Review History

RSOS-211121.R0 (Original submission)

Review form: Reviewer 1

Is the manuscript scientifically sound in its present form?

Yes

Are the interpretations and conclusions justified by the results?

Yes

Is the language acceptable?

Yes

Do you have any ethical concerns with this paper?

No

Have you any concerns about statistical analyses in this paper?

No

Recommendation?

Major revision is needed (please make suggestions in comments)

Comments to the Author(s)

Heesen et al. present naturalistic evidence for joint commitment in great apes through observation of spontaneous interruptions, similar to the experimental interruptions of their previous paper. The addition of chimpanzees and more ecologically meaningful setting is a valuable progression from previous work, and the size of the data set is commendable. There are a couple concerns, however, that should be more directly stated in the manuscript before it can be recommended for publication, along with some minor suggested revisions.

Major concerns:

The paper rests on their analysis of natural interruptions to joint behaviour, however the term interruption is not well-defined here and may be used in different ways. The authors focus in the introduction on external interruptions, where some event causes the joint action to be disrupted, however in their methods they define an interruption as any time the action stops. What differentiates an interruption from a natural end to the behaviour? Without clearly stating why some behaviours are interrupted and some come to a natural end, it seems any behaviour which continues after stopping for any duration of time (<2min) qualifies.

The other major concern is whether the chosen minimum amount of time (3s) to be considered a real interruption is meaningful. It seems with such a short period one could interpret the behaviour as continuing, especially if it was just a glance away. In my observation of great apes such pauses are relatively common, but often do not seem to be really interrupted. Was there a reason this threshold was chosen?

Other comments:

In the introduction or discussion, there should be some mention about whether non-social activities may elicit the same pattern. In the previous study, social activities were resumed at a higher rate following experimental interruptions, so this should be mentioned.

In many previous joint intentionality studies, there is an external task or goal, or the action is object oriented. Do you think this may form a different kind of joint actions and commitment? In the previous bonobo study, evidence was found for joint commitments on dyadic, purely socially directed behaviour. This possibility should at least be mentioned as opposed to concluding negative evidence is due to a lack of ecological validity. Perhaps dyadic social actions engage a different kind of joint commitment (worthy of testing in more ecologically valid situations in the future).

While bonobos are often thought to be more socially aware in some settings, chimpanzees more readily engage in some types of cooperative activities like group hunting and border patrols in the wild which may favour the evolution of joint commitment.

Did you ever see triadic play or grooming? If so how were these coded/analyzed? Do you predict a difference? It is not necessary to expand on this in detail in the discussion but it would be interesting to mention if you have the data.

Some mention of the evolutionary selection for joint commitment may be helpful in connecting to broader literature, not only demonstrating it exists in great apes. Human literature emphasizes, for example, collaborative extractive foraging. If the behavioural pattern emerged earlier in evolution, do you predict contexts or settings which select for it?

Line by line comments:

Line 105-106: I think this should read "bonobos are said to be more perceptive of..."

Lines 162-167: Some more details on how DSI are calculated would be helpful given how many measures are used.

Lines 186-194: this paragraph is a bit confusing as written as you go between resumptions, non resumption, and back to resumptions then non resumptions, using some slightly different language (a resumption vs have resumed).

Line 291-192: are the same play type and same grooming area meaningfully the same? In the context of action continuation as you put it there certainly is relevance that both were continued, but I wonder if absolute rates should be compared for these quite different behaviours.

Was model 2 run using all data or focused on data for which there was a resumption? It may be a stronger test if even within resumptions, it was more likely for the same action pattern/body area of grooming to be continued.

Line 317-318: In this sentence, does this imply you counted some behaviour as resumed even if they were with a different partner? This does not sound like a joint commitment if the partner changed but mere individual motivation to continue the behaviour. Since such a high proportion were indeed with the same partner, I would recommend focusing on these as much clearer evidence (if this was already done, please rephrase this sentence).

Line 359: should read "...with a slight tendency for bonobos.."

358-363: it might be worthwhile to mention that both species in the wild cooperate in different ways, especially that chimpanzees may be less socially tolerant but engage in more large scale cooperative behaviours

Figure 1: The caption of this figure should define action continuation (same body part/play type) and whether this represents data within behaviour resumptions or absolute rates. In both a and b, why is there a dotted line at 50%? I don't think this line is necessary as chance values here would not be resumming the exact same action half the time, it may give some readers the impression chimpanzees do not resume actions.

Review form: Reviewer 2

Is the manuscript scientifically sound in its present form?

Yes

Are the interpretations and conclusions justified by the results?

Yes

Is the language acceptable?

Yes

Do you have any ethical concerns with this paper?

No

Have you any concerns about statistical analyses in this paper?

No

Recommendation?

Accept with minor revision (please list in comments)

Comments to the Author(s)

See the attached file (Appendix A).

Decision letter (RSOS-211121.R0)

Dear Dr Heesen

The Editors assigned to your paper RSOS-211121 "Evidence of joint commitment in great ape natural joint actions" have now received comments from reviewers and would like you to revise the paper in accordance with the reviewer comments and any comments from the Editors. Please note this decision does not guarantee eventual acceptance.

Please submit your revised manuscript and required files (see below) no later than 21 days from today's (ie 16-Aug-2021) date. Note: the ScholarOne system will 'lock' if submission of the revision is attempted 21 or more days after the deadline. If you do not think you will be able to meet this deadline please contact the editorial office immediately.

on behalf of Dr Shinya Yamamoto (Associate Editor) and Kevin Padian (Subject Editor)
openscience@royalsociety.org

Associate Editor Comments to Author (Dr Shinya Yamamoto):

Reading your manuscript myself, I agree with the reviewers' comments, and believe these help you improve your manuscript. I look forward to reading your revision.

Reviewer comments to Author:

Reviewer: 1

Comments to the Author(s)

Heesen et al. present naturalistic evidence for joint commitment in great apes through observation of spontaneous interruptions, similar to the experimental interruptions of their previous paper. The addition of chimpanzees and more ecologically meaningful setting is a valuable progression from previous work, and the size of the data set is commendable. There are a couple concerns, however, that should be more directly stated in the manuscript before it can be recommended for publication, along with some minor suggested revisions.

Major concerns:

The paper rests on their analysis of natural interruptions to joint behaviour, however the term interruption is not well-defined here and may be used in different ways. The authors focus in the introduction on external interruptions, where some event causes the joint action to be disrupted, however in their methods they define an interruption as any time the action stops. What differentiates an interruption from a natural end to the behaviour? Without clearly stating why some behaviours are interrupted and some come to a natural end, it seems any behaviour which continues after stopping for any duration of time (<2min) qualifies.

The other major concern is whether the chosen minimum amount of time (3s) to be considered a real interruption is meaningful. It seems with such a short period one could interpret the behaviour as continuing, especially if it was just a glance away. In my observation of great apes such pauses are relatively common, but often do not seem to be really interrupted. Was there a reason this threshold was chosen?

Other comments:

In the introduction or discussion, there should be some mention about whether non-social activities may elicit the same pattern. In the previous study, social activities were resumed at a higher rate following experimental interruptions, so this should be mentioned.

In many previous joint intentionality studies, there is an external task or goal, or the action is object oriented. Do you think this may form a different kind of joint actions and commitment? In the previous bonobo study, evidence was found for joint commitments on dyadic, purely socially directed behaviour. This possibility should at least be mentioned as opposed to concluding negative evidence is due to a lack of ecological validity. Perhaps dyadic social actions engage a different kind of joint commitment (worthy of testing in more ecologically valid situations in the future).

While bonobos are often thought to be more socially aware in some settings, chimpanzees more readily engage in some types of cooperative activities like group hunting and border patrols in the wild which may favour the evolution of joint commitment.

Did you ever see triadic play or grooming? If so how were these coded/analyzed? Do you predict a difference? It is not necessary to expand on this in detail in the discussion but it would be interesting to mention if you have the data.

Some mention of the evolutionary selection for joint commitment may be helpful in connecting to broader literature, not only demonstrating it exists in great apes. Human literature emphasizes, for example, collaborative extractive foraging. If the behavioural pattern emerged earlier in evolution, do you predict contexts or settings which select for it?

Line by line comments:

Line 105-106: I think this should read "bonobos are said to be more perceptive of..."

Lines 162-167: Some more details on how DSI are calculated would be helpful given how many measures are used.

Lines 186-194: this paragraph is a bit confusing as written as you go between resumptions, non resumption, and back to resumptions then non resumptions, using some slightly different language (a resumption vs have resumed).

Line 291-192: are the same play type and same grooming area meaningfully the same? In the context of action continuation as you put it there certainly is relevance that both were continued, but I wonder if absolute rates should be compared for these quite different behaviours. Was model 2 run using all data or focused on data for which there was a resumption? It may be a stronger test if even within resumptions, it was more likely for the same action pattern/body area of grooming to be continued.

Line 317-318: In this sentence, does this imply you counted some behaviour as resumed even if they were with a different partner? This does not sound like a joint commitment if the partner changed but mere individual motivation to continue the behaviour. Since such a high proportion were indeed with the same partner, I would recommend focusing on these as much clearer evidence (if this was already done, please rephrase this sentence).

Line 359: should read "...with a slight tendency for bonobos.."

358-363: it might be worthwhile to mention that both species in the wild cooperate in different ways, especially that chimpanzees may be less socially tolerant but engage in more large scale cooperative behaviours

Figure 1: The caption of this figure should define action continuation (same body part/play type) and whether this represents data within behaviour resumptions or absolute rates. In both a and b, why is there a dotted line at 50%? I don't think this line is necessary as chance values here would not be resuming the exact same action half the time, it may give some readers the impression chimpanzees do not resume actions.

Reviewer: 2

Comments to the Author(s)

See the attached file ("RSOS_comments_to_Authors.pdf")

===PREPARING YOUR MANUSCRIPT===

If you have been asked to revise the written English in your submission as a condition of publication, you must do so, and you are expected to provide evidence that you have received language editing support. The journal would prefer that you use a professional language editing service and provide a certificate of editing, but a signed letter from a colleague who is a native

speaker of English is acceptable. Note the journal has arranged a number of discounts for authors using professional language editing services (<https://royalsociety.org/journals/authors/benefits/language-editing/>).

===PREPARING YOUR REVISION IN SCHOLARONE===

<https://royalsociety.org/journals/authors/author-guidelines/#supplementary-material> to include a suitable title and informative caption. An example of appropriate titling and captioning may be found at https://figshare.com/articles/Table_S2_from_Is_there_a_trade-

off_between_peak_performance_and_performance_breadth_across_temperatures_for_aerobic_sc
ope_in_teleost_fishes_/3843624.

Author's Response to Decision Letter for (RSOS-211121.R0)

See Appendices B & C.

RSOS-211121.R0 (Original submission)

Review form: Reviewer 1

Is the manuscript scientifically sound in its present form?

Yes

Are the interpretations and conclusions justified by the results?

Yes

Is the language acceptable?

Yes

Do you have any ethical concerns with this paper?

No

Have you any concerns about statistical analyses in this paper?

No

Recommendation?

Accept with minor revision (please list in comments)

Comments to the Author(s)

I thank the authors for an excellent revision. I am happy to see care has been taken to respond to mine and the other reviewer's comments, and for the most part satisfied the concerns I had raised. I am especially glad to see that many of my more serious concerns were due to unclarity in the text rather than problems in the methods and analysis, which have not been satisfactorily described. This is an interesting paper and I think it represents a significant improvement from the first draft and contribution to the literature, though remain some important points I think should be raised before I can fully endorse publication.

Most significantly, while the authors now clarify based on my previous review they analyzed only external interruptions as opposed to natural ends, this term was not operationalized clearly. Without a definition for this, it is impossible to evaluate the rigour of these observations. While in the introduction the authors say "we recorded the apes' behaviours following both internal causes (partner(s) taking a break) and external causes (partner(s) stopping the activity due to a noise, conspecific interrupting, or other kindred events) of interruptions," "kindred events" is not specific enough to evaluate as a definition. In the methods you state "stop all movements

characteristics of the activity, and produce at least one glance away from the activity" as part of the definition, but this would always be observed in a natural end as well (though I know this distinguishes between pause and interruption which I appreciate identifying). An operational definition of interruption compared to natural end is essential to be found in the methods.

I appreciate the very engaging and interesting discussion about triadic joint commitments in the discussion, though I think some mention of this should be included in the introduction as well when discussing what separates your study from others. While it is true that many previous studies may lack ecological validity, it is also a possibly important difference that they almost entirely focus on triadic tasks and thus may be unsuitable in that way (or at least leave an open question about whether great apes engage in dyadic joint commitment). I think some mention of this would help readers follow another key difference between yours and previous studies, as well as link more smoothly to the interesting discussion about it later on.

It is not necessary to include this in the manuscript, but have you tried analyzing the likelihood of resumption by social rank/social bonds based on timing? With the very short intervals, there may be much smaller likelihood of abandoning the activity in general, but as intervals become longer it will require more dedicated effort to resume a joint activity. Perhaps floor effects in the shortest intervals limit the rank/bond differences, but that greater differences would be more noticeable when looking at the longer intervals (or perhaps interruptions involving moving from the location and returning using your partner separation variable). In cases where it is noticeable, if it can be operationalized well, it may also be interesting to look at unilateral interruptions as compared to both partners getting distracted, similar to your earlier study involving experimental interruptions. There may not be sufficient data for this, but it will be an interesting question in the future. If the majority of cases came from distractions to both partners (which may be less salient than your earlier experimental interruptions which seem to have prompted suspension communication and high resumption) then there may be less importance of "politeness" or its equivalent since it is mutual, compared to one partner getting distracted and attempting to resume.

Thank you for including more details about DSI calculation. One question I have is how this calculation compensates for differing coresidency in the San Diego group you mention you performed? It is a bit confusing here as well and is an important point for social relations with differing grouping so more details here would be appreciated.

Regarding the choice of minimum time to count as an interruption, I appreciate the greater discussion, though I wonder how relevant the signalling response time is as the basis to set this definition. Waiting for a response after a signal seems a very different thing than time at which an interruption is recognized as an interruption. I would also recommend including another histogram with more bins, since the first bar in the histogram is far larger than any others, but you suggest only a small proportion of interruptions were ≤ 5 s. A histogram focusing perhaps on these first bars with many more bins would clarify that a bit more and be more convincing. In the paragraph starting on line 429 and elsewhere it would be good to mention again you mean specifically the same style of behaviour (location or play type) rather than just the general categories of behaviour (grooming, play). While you earlier emphasize this point of the analysis, for a naïve reader who does not focus on all the details it may be confusing. Play and grooming are themselves behaviours, so you should be clear you mean an even higher level of specificity. Another point that I do not think is necessary to include in the MS is whether the fact that social behaviours can be rewarding is an important difference from your solitary control in previous studies. If some rewarding individual behaviour is interrupted, they may resume more, perhaps the fact that social behaviour are rewarding, combined with recency bias, rather than pure commitment can be an alternative understanding which needs some controlled conditions to rule out.

The point about triadic joint commitment may fit into the final paragraph as well, since collaborative extractive foraging may depend more on this triadic joint commitment than dyadic. It is also worth noting that group hunting, female alliances attacking male bonobos, and chimpanzee border patrols may be more triadic than dyadic in some sense, even though the

attention target is not always in direct sight (prey animals, troublesome males, or outgroup members, respectively). The dyadic commitments of bonobos and chimpanzees you study may be closely related to social bond development/maintenance, coalition formation, and group stability/cohesion, while triadic joint commitment warrants future study based on these behaviours observed in chimpanzees and bonobos.

Another question is whether you ever saw play or grooming in groups? I often have seen chimpanzees play with more than one individual and regularly see bonobos in grooming parties with more than 2 individuals, were these ever observed in your groups? If so, how were they coded and analyzed?

While both species hunt monkeys, and there is likely under reporting of the frequency in bonobos, I think it should be pointed out that there appears to be a species difference here.

Chimpanzees have been observed hunting more frequently, and in one of the bonobo papers you cite (Hohmann and Fruth 2008) there is evidence for meat eating but no evidence for collaborative hunting presented. There are several bonobo groups which have been observed daily for years but the frequency of observation of group hunting is far lower, never reported at some sites. The prey choice also differs, where bonobos tend to prefer individual hunting of duikers, chimpanzees prefer group hunting of monkeys (though these are of course far from exclusive tendencies). Bonobos have not been observed hunting red colobus monkeys, the preferred prey of chimpanzees, but instead have been observed mutual grooming with them. The species seem to have different kinds of social intelligence here, and their collaborative hunting should not be lumped together without at least a disclaimer that it is seen more rarely in bonobos. A supplementary video of an example of short interruption would be beneficial, currently all videos show long interruptions, which although possibly the most interesting are not representative of all your observations, so some examples on the shorter end should be included.

I thank the authors for their care in responding and an excellent revision.

Review form: Reviewer 2

Is the manuscript scientifically sound in its present form?

Yes

Are the interpretations and conclusions justified by the results?

Yes

Is the language acceptable?

Yes

Do you have any ethical concerns with this paper?

No

Have you any concerns about statistical analyses in this paper?

No

Recommendation?

Accept as is

Comments to the Author(s)

I think that the authors did a good job clarifying the operationalizations and enriching the discussion, which now more clearly and explicitly communicates the contribution of the current work to the field.

I did spot a small typo (Line 436: "evinced" should be evidenced) that will need to get fixed at some point during the process.

Decision letter (RSOS-211121.R1)

Dear Dr Heesen,

On behalf of the Editors, we are pleased to inform you that your Manuscript RSOS-211121.R1 "Evidence of joint commitment in great apes' natural joint actions" has been accepted for publication in Royal Society Open Science subject to minor revision in accordance with the referees' reports. Please find the referees' comments along with any feedback from the Editors below my signature.

Please submit your revised manuscript and required files (see below) no later than 7 days from today's (ie 04-Oct-2021) date. Note: the ScholarOne system will 'lock' if submission of the revision is attempted 7 or more days after the deadline. If you do not think you will be able to meet this deadline please contact the editorial office immediately.

on behalf of Dr Shinya Yamamoto (Associate Editor) and Kevin Padian (Subject Editor)
openscience@royalsociety.org

Associate Editor Comments to Author (Dr Shinya Yamamoto):

I acknowledge that the authors have improved their manuscript considerably, and now it is close to acceptance for publication. However, the reviewer 1 still raised some important points. Please take his/her comments fully into consideration.

Reviewer comments to Author:

Reviewer: 1

Comments to the Author(s)

I thank the authors for an excellent revision. I am happy to see care has been taken to respond to mine and the other reviewer's comments, and for the most part satisfied the concerns I had raised. I am especially glad to see that many of my more serious concerns were due to unclarity in the text rather than problems in the methods and analysis, which have not been satisfactorily described. This is an interesting paper and I think it represents a significant improvement from the first draft and contribution to the literature, though remain some important points I think should be raised before I can fully endorse publication.

Most significantly, while the authors now clarify based on my previous review they analyzed only external interruptions as opposed to natural ends, this term was not operationalized clearly. Without a definition for this, it is impossible to evaluate the rigour of these observations. While in the introduction the authors say "we recorded the apes' behaviours following both internal causes (partner(s) taking a break) and external causes (partner(s) stopping the activity due to a noise, conspecific interrupting, or other kindred events) of interruptions," "kindred events" is not specific enough to evaluate as a definition. In the methods you state "stop all movements characteristics of the activity, and produce at least one glance away from the activity" as part of the definition, but this would always be observed in a natural end as well (though I know this distinguishes between pause and interruption which I appreciate identifying). An operational definition of interruption compared to natural end is essential to be found in the methods.

I appreciate the very engaging and interesting discussion about triadic joint commitments in the discussion, though I think some mention of this should be included in the introduction as well when discussing what separates your study from others. While it is true that many previous studies may lack ecological validity, it is also a possibly important difference that they almost entirely focus on triadic tasks and thus may be unsuitable in that way (or at least leave an open question about whether great apes engage in dyadic joint commitment). I think some mention of this would help readers follow another key difference between yours and previous studies, as well as link more smoothly to the interesting discussion about it later on.

It is not necessary to include this in the manuscript, but have you tried analyzing the likelihood of resumption by social rank/social bonds based on timing? With the very short intervals, there may be much smaller likelihood of abandoning the activity in general, but as intervals become longer it will require more dedicated effort to resume a joint activity. Perhaps floor effects in the shortest intervals limit the rank/bond differences, but that greater differences would be more noticeable when looking at the longer intervals (or perhaps interruptions involving moving from the location and returning using your partner separation variable). In cases where it is noticeable, if it can be operationalized well, it may also be interesting to look at unilateral interruptions as compared to both partners getting distracted, similar to your earlier study involving experimental interruptions. There may not be sufficient data for this, but it will be an interesting question in the future. If the majority of cases came from distractions to both partners (which may be less salient than your earlier experimental interruptions which seem to have prompted suspension communication and high resumption) then there may be less importance of "politeness" or its equivalent since it is mutual, compared to one partner getting distracted and attempting to resume.

Thank you for including more details about DSI calculation. One question I have is how this calculation compensates for differing coresidency in the San Diego group you mention you performed? It is a bit confusing here as well and is an important point for social relations with differing grouping so more details here would be appreciated.

Regarding the choice of minimum time to count as an interruption, I appreciate the greater discussion, though I wonder how relevant the signalling response time is as the basis to set this definition. Waiting for a response after a signal seems a very different thing than time at which an interruption is recognized as an interruption. I would also recommend including another histogram with more bins, since the first bar in the histogram is far larger than any others, but you suggest only a small proportion of interruptions were <5s. A histogram focusing perhaps on these first bars with many more bins would clarify that a bit more and be more convincing. In the paragraph starting on line 429 and elsewhere it would be good to mention again you mean specifically the same style of behaviour (location or play type) rather than just the general categories of behaviour (grooming, play). While you earlier emphasize this point of the analysis, for a naïve reader who does not focus on all the details it may be confusing. Play and grooming are themselves behaviours, so you should be clear you mean an even higher level of specificity.

Another point that I do not think is necessary to include in the MS is whether the fact that social behaviours can be rewarding is an important difference from your solitary control in previous studies. If some rewarding individual behaviour is interrupted, they may resume more, perhaps the fact that social behaviour are rewarding, combined with recency bias, rather than pure commitment can be an alternative understanding which needs some controlled conditions to rule out.

The point about triadic joint commitment may fit into the final paragraph as well, since collaborative extractive foraging may depend more on this triadic joint commitment than dyadic. It is also worth noting that group hunting, female alliances attacking male bonobos, and chimpanzee border patrols may be more triadic than dyadic in some sense, even though the attention target is not always in direct sight (prey animals, troublesome males, or outgroup members, respectively). The dyadic commitments of bonobos and chimpanzees you study may be closely related to social bond development/maintenance, coalition formation, and group stability/cohesion, while triadic joint commitment warrants future study based on these behaviours observed in chimpanzees and bonobos.

Another question is whether you ever saw play or grooming in groups? I often have seen chimpanzees play with more than one individual and regularly see bonobos in grooming parties with more than 2 individuals, were these ever observed in your groups? If so, how were they coded and analyzed?

While both species hunt monkeys, and there is likely under reporting of the frequency in bonobos, I think it should be pointed out that there appears to be a species difference here. Chimpanzees have been observed hunting more frequently, and in one of the bonobo papers you cite (Hohmann and Fruth 2008) there is evidence for meat eating but no evidence for collaborative hunting presented. There are several bonobo groups which have been observed daily for years but the frequency of observation of group hunting is far lower, never reported at some sites. The prey choice also differs, where bonobos tend to prefer individual hunting of duikers, chimpanzees prefer group hunting of monkeys (though these are of course far from exclusive tendencies). Bonobos have not been observed hunting red colobus monkeys, the preferred prey of chimpanzees, but instead have been observed mutual grooming with them. The species seem to have different kinds of social intelligence here, and their collaborative hunting should not be lumped together without at least a disclaimer that it is seen more rarely in bonobos.

A supplementary video of an example of short interruption would be beneficial, currently all videos show long interruptions, which although possibly the most interesting are not representative of all your observations, so some examples on the shorter end should be included.

I thank the authors for their care in responding and an excellent revision.

Reviewer: 2

Comments to the Author(s)

I think that the authors did a good job clarifying the operationalizations and enriching the discussion, which now more clearly and explicitly communicates the contribution of the current work to the field.

I did spot a small typo (Line 436: "evinced" should be evidenced) that will need to get fixed at some point during the process.

===PREPARING YOUR MANUSCRIPT===

===PREPARING YOUR REVISION IN SCHOLARONE===

Author's Response to Decision Letter for (RSOS-211121.R1)

See Appendix D.

Decision letter (RSOS-211121.R2)

Dear Dr Heesen,

I am pleased to inform you that your manuscript entitled "Evidence of joint commitment in great apes' natural joint actions" is now accepted for publication in Royal Society Open Science.

on behalf of Dr Shinya Yamamoto (Associate Editor) and Kevin Padian (Subject Editor)
openscience@royalsociety.org

Appendix A

The current paper is essentially the field study counterpart of an experimental study in an earlier paper (Heesen et al., 2020) addressing the same question. I think the data in the current study complements that previous experimental study well, replicating some of the findings in an ecologically valid setting while also providing some new insights. As such, I think the data is a valuable addition to the discussion on mental and behavioral jointness in non-human animals. However, I do believe that the current results need to be discussed much more explicitly and elaborately by the authors, in order to fully communicate their contribution to the literature.

1. In general, I would argue that the Heesen et al., (2020) makes several important interpretative points (e.g., with regard to alternative explanations) that would also apply to the current data, but are missing in the current manuscript. For example, readers might still argue for completion effects or the “novel interaction” explanation. The current results need to be discussed with regards to these issues, if necessary combining the current results with those of the Heesen et al., (2020).
2. One particular area that is briefly touched on in the Heesen et al., (2020) paper, but not in the current manuscript, is the degree to which the apes’ commitment is characterized by assumptions of mutually inferred mental states, or common ground about the jointness of the commitment.

A discussion of this issue seems particularly pertinent given recent comparative studies on joint attention and social bonding. These studies showed that, like 2.5 year old children, merely attending to a video together caused apes to be more likely to seek out social interaction afterwards. This suggests that apes’ behavior is indeed sensitive to the degree of overlap between their mental states and that of their social partners (Wolf & Tomasello, 2019, 2020a).

However, subsequent research also showed that, in contrast to human children, the apes were not sensitive to others’ attempts to create a mutual understanding or common ground about their experience being shared (Wolf & Tomasello, 2020b). This difference, albeit subtle, is crucial with regard to how the cognitive machinery underlying this social behavior is conceptualized in the current manuscript, especially with regard to the question whether this behavior is driven by a notion of obligation. Do great apes perhaps merely infer when others’ goals overlap with theirs, and thus seek out to resume social interaction (and the initial activity) until they deem it unlikely that their and their partner’s motivations still overlap? Some scholars and philosophers would argue that this psychological mechanism, albeit a precursor, is not truly a sense of obligation in the human sense of the word, because the individual never constitutes a ‘we’, nor prioritizes the ‘we’ over the ‘me’ the way that humans do (e.g., Tomasello, 2020).

Or do in fact apes conceptualize joint mental states and joint action in a more mutualistic sense? Is their reengagement following interruption not only motivated by the inferred overlap of goals with their partner, but also by assumptions about whether the partner expects these inferences and assumptions to be mutual, meaning that they conceptualize their activity truly jointly, as a ‘we’?

3. Another issue that needs to be discussed in more detail is that the social engagement of the apes in the current study was beneficial/psychologically rewarding. Yet, one crucial element of human joint commitment is that it persists even beyond a motivation for attaining individual rewards. For example: in the Hamann et al. (2012) study, 3.5 year olds helped a collaborative partner getting their reward even after they had already received theirs. This suggests a true sense of we-conceptualization, resulting in a sense of obligation: although I am finished with the task (i.e., I got all my rewards), we did not finish yet (i.e., we did not get all our rewards). To me, it seems unlikely that apes' joint commitment reaches that far (perhaps suggesting that apes' conceptualization of the joint commitment, for example in terms of common ground, might still be different than that of humans), although the authors are free to argue otherwise, as long as this point of individual vs collective (psychological) rewards within joint action is discussed.

4. One additional thing I am somewhat unclear about is the issue of politeness, which seems not be completely fleshed out.

The authors wonder whether resuming a joint action can be construed as an act of politeness in apes. In humans, politeness in discourse can be easily recognized by specific patterns in interactions (either verbally or non-verbally) as they are often culturally conventionalized. However, if one only uses the rate of joint action resumptions following interruptions to infer a sense of politeness, this becomes much more difficult, as similar behavior would emerge from a sense of social closeness as well. This makes it very difficult to assess whether such behavior is motivated by politeness or by a desire to interact stemming from a previously established social bond with a partner. Surely, in many contexts, humans would resume joint actions after an interruption more often with a friend than with a stranger, even in cultures where politeness is considered important? In other words, I am not quite clear on how, in this study, the authors distinguished between resuming a joint commitment out of politeness and resuming a joint commitment motivated by a previously existing strong social bond (i.e., just wanting to resume doing things with a friend). Are these two qualitatively distinguishable in terms of behavior? If so, this is not clear from the manuscript. Nor am I clear how this rationale is used in combination with the data to make inferences about politeness heuristics.

As such, I think that the politeness argument needs to be fleshed out considerably more, not only in the general rationale of the argument, but also specifically in terms of how this was measured in the study, and distinguished from others forms of behavior: from which observable behavior in which context do the the authors infer a sense of politeness, and why? And how would we know this is indeed politeness we are seeing in not something else (e.g., behavior motivated by friendship/alliance membership).

5. Do the authors have any idea why bonobos were more likely to continue their specific activity compared to Chimpanzees?

Minor issues

- It might be helpful for the reader to change the labels of action resumption to be more intuitively distinguishable from the action continuation. As the resumption variable is really about resuming the social interaction rather than the previous behavior specifically, would it be an idea to change this into 'social interaction resumption' (vs 'activity continuity')? Also, given that these are 2 crucial independent variables, it might help explicitly reiterating their precise nature again while describing the statistical models and/or in the results.
- Although I quite like figures 1, I am wondering if the reference line at $p = .5$ is really necessary? There is no baseline probability of .5 and the line somewhat interferes with some of the datapoints (in particular with those of the chimpanzees).

Literature:

- Hamann, K., Warneken, F., & Tomasello, M. (2012). Children's Developing Commitments to Joint Goals: Children's Developing Commitments to Joint Goals. *Child Development, 83*(1), 137–145. <https://doi.org/10.1111/j.1467-8624.2011.01695.x>
- Tomasello, M., & Hamann, K. (2012). Collaboration in Young Children. *Quarterly Journal of Experimental Psychology, 65*(1), 1–12. <https://doi.org/10.1080/17470218.2011.608853>
- Wolf, W., & Tomasello, M. (2019). Visually attending to a video together facilitates great ape social closeness. *Proceedings of the Royal Society B, 286*(19087).
- Wolf, W., & Tomasello, M. (2020a). Watching a video together creates social closeness between children and adults. *Journal of Experimental Child Psychology, 189*, 12.
- Wolf, W., & Tomasello, M. (2020b). Human children, but not great apes, become socially closer by sharing an experience in common ground. *Journal of Experimental Child Psychology, 199*, 104930. <https://doi.org/10.1016/j.jecp.2020.104930>

Appendix B

Dr. Raphaela Heesen
Department of Psychology
Durham University, UK
South Rd, Durham
DH1 3LE, UK
Phone: +44 (0) 7389 777274

21st September 2021

Dear Editor,

We would like to thank you for the opportunity to revise our manuscript (RSOS-211121), now entitled "Evidence of joint commitment in great apes' natural joint actions". We found the reviewers' comments to be very helpful in revising our manuscript and have carefully considered and responded to each of their concerns.

Following the editorial and reviewer queries, we have proceeded to textual changes in the introduction, methods and discussion, in order to improve clarity and aid comprehension of our findings as well as to justify our choices with respect to interruption definitions (requests by reviewer 1). We have also moved a supplementary figure (now Figure 1) into the results, to provide more details on the descriptive statistics of interruptions. Furthermore, we addressed – as suggestion by reviewer 1 and 2 – several important questions related to the broader significance of our findings. Notably, we have added paragraphs to our discussion, in which we place our research findings into the broader context of the literature, discussing these in light with the current theories on human cooperation.

We have attached a "Response to Reviewers" in which we address each reviewer's comment in detail; we have complied with all of the reviewers' comments. All corresponding changes made to the revised manuscript are highlighted in yellow (track change version) and referenced in our response to reviewers.

We believe the manuscript has been substantially improved as a result of the reviewers' constructive comments, and we very much hope that our revised version is now acceptable for publication in *Royal Society Open Science*.

Thank you for your consideration, we look forward to hearing from you in due course.

Sincerely,

Raphaela Heesen (and co-authors)

Appendix C

Response to reviews

Editorial decision: Major revision within 21 days

We are grateful to the editors and reviewers for their time and expertise in reviewing this manuscript. We have now revised the manuscript in accordance with the reviewers' constructive comments and responded to each concern, which we detail below. Overall, we believe the revisions have resulted in a more nuanced and improved manuscript that makes a much broader contribution to the literature. References used in our response are provided in a bibliography at the end of this letter.

Reviewer comments to Author:

Reviewer: 1

Comments to the Author(s)

Heesen et al. present naturalistic evidence for joint commitment in great apes through observation of spontaneous interruptions, similar to the experimental interruptions of their previous paper. The addition of chimpanzees and more ecologically meaningful setting is a valuable progression from previous work, and the size of the data set is commendable. There are a couple concerns, however, that should be more directly stated in the manuscript before it can be recommended for publication, along with some minor suggested revisions.

Major concerns:

The paper rests on their analysis of natural interruptions to joint behaviour, however the term interruption is not well-defined here and may be used in different ways. The authors focus in the introduction on external interruptions, where some event causes the joint action to be disrupted, however in their methods they define an interruption as any time the action stops. What differentiates an interruption from a natural end to the behaviour? Without clearly stating why some behaviours are interrupted and some come to a natural end, it seems any behaviour which continues after stopping for any duration of time (<2min) qualifies.

We thank the reviewer for raising this important point. First, to match our methods with the introduction more consistently, we now clarify that – although we video recorded all interruptions (internal and external causes), we only analysed interruptions caused by *external* events (see introduction LL94-98). The reason for this decision was precisely due to the point the reviewer is raising here, namely that with interruptions of internal causes (one or both partners suddenly stopping the interaction) is it less clear whether this represents an interruption or natural ending. The fact that bonobos frequently continued the joint action and behaviour with the same partner, and at the same location (even after having moved away from the original location), does however suggest that we are observing resumptions of interrupted joint actions rather than initiation of new ones. To further answer the point with regards to the interruption definition - the data actually shows that the mean duration of interruptions was 22.7 sec (SD = 32.3 sec, 1st quartile = 5.6 sec; 3rd quartile: 23.5 sec), and interruptions were rarely above 150 sec (1.3 %). Thus, our maximum time frame of 2 minutes is reasonable as it represents the far end of the duration spectrum. We have now added this information to our results (LL 348-352) and added a histogram of interruption durations to the Supplementary Materials to demonstrate the point (Fig. S3, see below). Additionally, we moved the descriptive figure on interruption durations (originally Fig. S3) into the main results (now Fig. 1).

Figure S3 (reprinted from manuscript supplementary materials). Histogram on interruption durations (in seconds) including all interruptions after which joint actions were resumed ($N = 2,455$). Note that interruptions of duration > 200 seconds denote interruptions where one or both partners visibly reengaged their original partner (see for criteria methods page 5, section “Video coding”).

The other major concern is whether the chosen minimum amount of time (3s) to be considered a real interruption is meaningful. It seems with such a short period one could interpret the behaviour as continuing, especially if it was just a glance away. In my observation of great apes such pauses are relatively common, but often do not seem to be really interrupted. Was there a reason this threshold was chosen?

We operationalized the minimum duration of an interruption based on the observation that 3 sec is a minimal time window required for an individual to stop all movements characteristics of the activity, and produce at least one glance away from the activity. Considering that previous studies have used very minimal criteria (<1 sec to 5sec) of identifying ape response waiting after signalling (Graham, Furuichi, and Byrne, 2017; Hobaiter and Byrne, 2011; Rossano, 2013, Liebal, Call, and Tomasello, 2004) , our criteria of 3 sec is a reasonable time window. In any case, we would like to note that only a very small percentage (16%) of interruptions were less than 5 seconds of length (see Figure S3), indicating that observations of such short interruptions are anyways not highly representative of the data. A justification has now been added to our methods (LL 218-224).

Other comments:

In the introduction or discussion, there should be some mention about whether non-social activities may elicit the same pattern. In the previous study, social activities were resumed at a higher rate following experimental interruptions, so this should be mentioned.

Thanks for this suggestion. This is now mentioned in the discussion (LL 435-439).

In many previous joint intentionality studies, there is an external task or goal, or the action is object oriented. Do you think this may form a different kind of joint actions and commitment? In the

previous bonobo study, evidence was found for joint commitments on dyadic, purely socially directed behaviour. This possibility should at least be mentioned as opposed to concluding negative evidence is due to a lack of ecological validity. Perhaps dyadic social actions engage a different kind of joint commitment (worthy of testing in more ecologically valid situations in the future).

We thank the reviewer for this comment, and we agree that joint commitment studies in humans are often focused on triadic interactions (e.g., Gräfenhain, Behne, Carpenter, and Tomasello, 2009; Gräfenhain, Carpenter, and Tomasello, 2013; Kachel, Svetlova, and Tomasello, 2019), with a joint object of attention, or a clear perceivable goal, and as such, joint goals are more easily identifiable and explicit. Nonetheless, many studies in the human interaction literature equally study joint commitments in dyadic interactions of humans, such as conversation (e.g., Bangerter, Chevalley, and Derouwau, 2010; Chevalley and Bangerter, 2010; Clark, 2006; Mayor and Bangerter, 2015). Indeed, human conversation represents a good example of dyadic interactions that involve joint commitment. There is currently no evidence that the feeling of mutual obligation differs between triadic and dyadic interactions; however, the degree of coordination required may differ depending on the interaction nature and partner (meeting your boss vs. your friend, playing a ball game vs. having a simple chat).

We have now added a paragraph on this in our discussion (LL 439-449).

While bonobos are often thought to be more socially aware in some settings, chimpanzees more readily engage in some types of cooperative activities like group hunting and border patrols in the wild which may favour the evolution of joint commitment.

We thank the reviewer for raising this interesting point - we agree that complex collaborative activities might favour the evolution of joint commitment, but we believe that chimpanzees and bonobos have equal opportunities to engage in such collaborative activities. First of all, the collaborative nature of chimpanzees group hunting has only been described in one community and it is very much debated. Some argue that they rather engage in a parallel activity where each individual is following their individual and egoistic goal that is aligned with that of the others, and the same could be argued for border patrolling, which might not require a high level of coordination. Second, bonobos also engage in group hunting (Hohmann and Fruth, 2008; Surbeck and Hohmann, 2008) and cooperative aggression against out-group individuals (Tokuyama and Furuichi, 2016; Tokuyama, Sakamaki, and Furuichi, 2019). Third, close proximity and potentially risky interactions (due to close body contact) like grooming and play, in which both species engage in on a daily basis, might require more coordination of individual actions and communication. So we would argue that chimpanzees and bonobos do have equal opportunities to engage in cooperative activities and evolve joint commitment. Our result seem to support this assumption since bonobos and chimpanzees steadily resume social activities with previous partners, which suggests that both species share the ability to engage in joint commitment. With regards to species difference, we hypothesized that it would lie in the *quality* of joint commitment. We predicted that, given bonobos' heightened levels of social attention (Kano, Hirata, and Call, 2015), empathy and pro-sociality (Clay and de Waal, 2013; Tan, Ariely, and Hare, 2017), emotionality (Kret, Jaasma, Bionda, and Wijnen, 2016) and documented social tolerance/cognition (Hare, Melis, Woods, Hastings, and Wrangham, 2007; Herrmann, Hare, Call, and Tomasello, 2010), their joint commitments would be more fine-tuned to social partners as well as more evident in individuals' motivation to reconstruct previous behaviours (e.g., resuming to groom the same body part as before) – a prediction which indeed has partly been met with our data.

We have now added a passage in the discussion to discuss this issue (LL 485-500).

Did you ever see triadic play or grooming? If so how were these coded/analyzed? Do you predict a difference? It is not necessary to expand on this in detail in the discussion but it would be interesting to mention if you have the data.

We almost never observed triadic play or grooming. This type of triadic interactions are actually extremely rare in the natural behaviour of these species. We have a few examples of play interactions where an object was involved (like a stick for instance), but the object never seemed to be the shared focus of attention, and in most cases it was just held by one of the partners or they were just trying to steal it from the other but they never jointly played with the object. We now discuss the point on triadic interactions in the discussion (LL 439-449).

Some mention of the evolutionary selection for joint commitment may be helpful in connecting to broader literature, not only demonstrating it exists in great apes. Human literature emphasizes, for example, collaborative extractive foraging. If the behavioural pattern emerged earlier in evolution, do you predict contexts or settings which select for it?

We agree with the reviewer and we have now added mention on the evolutionary selection for joint commitment in the discussion (LL 485-500).

Line by line comments:

Line 105-106: I think this should read “bonobos are said to be more perceptive of...”

This has been corrected.

Lines 162-167: Some more details on how DSI are calculated would be helpful given how many measures are used.

More information on how the DSI are calculated is now provided in our method section (LL 197-199).

Lines 186-194: this paragraph is a bit confusing as written as you go between resumptions, non resumption, and back to resumptions then non resumptions, using some slightly different language (a resumption vs have resumed).

This has now been made clearer following the reviewer’s suggestion (LL 226-233).

Line 291-192: are the same play type and same grooming area meaningfully the same? In the context of action continuation as you put it there certainly is relevance that both were continued, but I wonder if absolute rates should be compared for these quite different behaviours.

This is an interesting point. However, as our research design entailed the comparison between different activity types from the start, we feel removing this comparison in our statistical models post hoc is not appropriate. To acknowledge the reviewer’s concern, we nonetheless indicated this limitation in our discussion (LL 427-432).

Was model 2 run using all data or focused on data for which there was a resumption? It may be a stronger test if even within resumptions, it was more likely for the same action pattern/body area of grooming to be continued.

Model 2 was run to answer the following research question: *when apes resume joint actions, are they motivated to continue the same action as before the interruption?* This question only focused on resumption data. To make this more clear to the reader, we have now added a sub-phrase in our introduction (L 104).

Line 317-318: In this sentence, does this imply you counted some behaviour as resumed even if they were with a different partner? This does not sound like a joint commitment if the partner changed but mere individual motivation to continue the behaviour. Since such a high proportion were indeed with the same partner, I would recommend focusing on these as much clearer evidence (if this was already done, please rephrase this sentence).

We now clarified that we only counted resumptions if resumed with the *same* interaction partner.

Line 359: should read "...with a slight tendency for bonobos.."

This has been corrected.

358-363: it might be worthwhile to mention that both species in the wild cooperate in different ways, especially that chimpanzees may be less socially tolerant but engage in more large scale cooperative behaviours

We agree with the reviewer and have now mentioned this in our discussion (LL 485-500), also citing relevant and current work on this matter.

Figure 1: The caption of this figure should define action continuation (same body part/play type) and whether this represents data within behaviour resumptions or absolute rates. In both a and b, why is there a dotted line at 50%? I don't think this line is necessary as chance values here would not be resuming the exact same action half the time, it may give some readers the impression chimpanzees do not resume actions.

We have now updated the Figure 1 a and b (now Figure 2 a and b) following the reviewer's suggestion and updated the R code on the online repository; we removed the 50% line and added the required information of whether these data involve resumptions to the caption. The parentheses (same body part/play type) inflated the size of the y-axis label, which is why we included it in the caption of the Figure instead.

Reviewer: 2

Comments to the Author(s)

The current paper is essentially the field study counterpart of an experimental study in an earlier paper (Heesen et al., 2020) addressing the same question. I think the data in the current study complements that previous experimental study well, replicating some of the findings in an ecologically valid setting while also providing some new insights. As such, I think the data is a valuable addition to the discussion on mental and behavioral jointness in non-human animals. However, I do believe that the current results need to be discussed much more explicitly and the elaborately by the authors, in order to fully communicate their contribution to the literature.

1. In general, I would argue that the Heesen et al., (2020) makes several important interpretative points (e.g., with regard to alternative explanations) that would also apply to the current data, but are missing in the current manuscript. For example, readers might still argue for completion effects or the "novel interaction" explanation. The current results need to be discussed with regards to these issues, if necessary combining the current results with those of the Heesen et al., (2020).

We thank the reviewer for raising this point and agree that this would improve our discussion. We have now addressed this issue and combined the new results with that of Heesen et al. (2020)'s study in the discussion (LL 474-485).

2. One particular area that is briefly touched on in the Heesen et al., (2020) paper, but not in the current manuscript, is the degree to which the apes' commitment is characterized by assumptions of mutually inferred mental states, or common ground about the jointness of the commitment. A discussion of this issue seems particularly pertinent given recent comparative studies on joint attention and social bonding. These studies showed that, like 2.5 year old children, merely attending to a video together caused apes to be more likely to seek out social interaction afterwards. This suggests that apes' behavior is indeed sensitive to the degree of overlap between their mental states and that of their social partners (Wolf & Tomasello, 2019, 2020a). However, subsequent research also showed that, in contrast to human children, the apes were not sensitive to others' attempts to create a mutual understanding or common ground about their experience being shared (Wolf & Tomasello, 2020b). This difference, albeit subtle, is crucial with regard to how the cognitive machinery underlying this social behavior is conceptualized in the current manuscript, especially with regard to the question whether this behavior is driven by a notion of obligation. Do great apes perhaps merely infer when others' goals overlap with theirs, and thus seek out to resume social interaction (and the initial activity) until they deem it unlikely that their and their partner's motivations still overlap? Some scholars and philosophers would argue that this psychological mechanism, albeit a precursor, is not truly a sense of obligation in the human sense of the word, because the individual never constitutes a 'we', nor prioritizes the 'we' over the 'me' the way that humans do (e.g., Tomasello, 2020). Or do in fact apes conceptualize joint mental states and joint action in a more mutualistic sense? Is their reengagement following interruption not only motivated by the inferred overlap of goals with their partner, but also by assumptions about whether the partner expects these inferences and assumptions to be mutual, meaning that they conceptualize their activity truly jointly, as a 'we'?

We thank the reviewer for this insightful comment –Following the reviewer's suggestion, we have now included a paragraph in our discussion to address this point (LL 450-473), along with the relevant literature mentioned by the reviewer. We believe this comment has significantly improved the quality of our discussion.

3. Another issue that needs to be discussed in more detail is that the social engagement of the apes in the current study was beneficial/psychologically rewarding. Yet, one crucial element of human joint commitment is that it persists even beyond a motivation for attaining individual rewards. For example: in the Hamann et al. (2012) study, 3.5 year olds helped a collaborative partner getting their reward even after they had already received theirs. This suggests a true sense of we-conceptualization, resulting in a sense of obligation: although I am finished with the task (i.e., I got all my rewards), we did not finish yet (i.e., we did not get all our rewards). To me, it seems unlikely that apes' joint commitment reaches that far (perhaps suggesting that apes' conceptualization of the joint commitment, for example in terms of common ground, might still be different than that of humans), although the authors are free to argue otherwise, as long as this point of individual vs collective (psychological) rewards within joint action is discussed.

We have now added a paragraph in the discussion to address this interesting comment in the discussion (LL 432-439).

4. One additional thing I am somewhat unclear about is the issue of politeness, which seems not be completely fleshed out.

The authors wonder whether resuming a joint action can be construed as an act of politeness in apes. In humans, politeness in discourse can be easily recognized by specific patterns in interactions (either verbally or non-verbally) as they are often culturally conventionalized. However, if one only uses the rate of joint action resumptions following interruptions to infer a sense of politeness, this becomes much more difficult, as similar behavior would emerge from a sense of social closeness as well. This makes it very difficult

to assess whether such behavior is motivated by politeness or by a desire to interact stemming from a previously established social bond with a partner. Surely, in many contexts, humans would resume joint actions after an interruption more often with a friend than with a stranger, even in cultures where politeness is considered important? In other words, I am not quite clear on how, in this study, the authors distinguished between resuming a joint commitment out of politeness and resuming a joint commitment motivated by a previously existing strong social bond (i.e., just wanting to resume doing things with a friend). Are these two qualitatively distinguishable in terms of behavior? If so, this is not clear from the manuscript. Nor am I clear how this rationale is used in combination with the data to make inferences about politeness heuristics.

As such, I think that the politeness argument needs to be fleshed out considerably more, not only in the general rationale of the argument, but also specifically in terms of how this was measured in the study, and distinguished from others forms of behavior: from which observable behavior in which context do the the authors infer a sense of politeness, and why? And how would we know this is indeed politeness we are seeing in not something else (e.g., behavior motivated by friendship/alliance membership).

We thank the reviewer for this important comment. We agree that in order to shed light on the evolutionary building blocks of politeness we should investigate the effects of social dimensions (friendship and power) on the communicative or behavioural processes involved in great ape joint action coordination. Two of our recent publications have actually evidenced that the communicative efforts deployed by bonobos to initiate, suspend, resume and terminate joint activities follow some of the patterns predicted by politeness theory (Heesen et al., 2021; Heesen, Bangerter, Zuberbühler, Iglesias, Rossano, Guéry, and Genty, 2020). Given that, in the present study, we only looked at resumption behaviour (not the specific communication efforts deployed to coordinate the suspension and resumption of the joint activity), we agree that we should avoid making assumptions regarding politeness but rather focus on the relative strength of commitments. Indeed, as suggested by the reviewer, the *strength of the commitment* or the motivation to resume the interrupted activity might differ depending on the relationship between the partners, probably in anticipation of eventual behavioural and social consequences, We have now adjusted the formulations in the corresponding paragraphs of our introduction (LL 114-123) and discussion (LL 400-405) accordingly.

5. Do the authors have any idea why bonobos were more likely to continue their specific activity compared to Chimpanzees?

Similar as to why bonobos are more tuned towards conspecifics' gaze (Kano, Hirata, and Call, 2015), exhibit attentional biases towards conspecifics' social emotions (Kret, Jaasma, Bionda, and Wijnen, 2016) or outcompete chimpanzees via social tolerance (Hare, Melis, Woods, Hastings, and Wrangham, 2007), we believe that increased continuation of behaviour rates in bonobos' grooming activities compared to that of chimpanzees' are a consequence of bonobo sociality and their high levels of partner-sensitivity/ social awareness. We have highlighted this explanation (as a justification of our prediction in the first place) in our introduction (LL 124-129).

Minor issues

It might be helpful for the reader to change the labels of action resumption to be more intuitively distinguishable from the action continuation. As the resumption variable is really about resuming the social interaction rather than the previous behaviour specifically, would it be an idea to change this into 'social interaction resumption' (vs 'activity continuity')? Also, given that these are 2 crucial independent variables, it might help explicitly reiterating their precise nature again while describing the statistical models and/or in the results.

We thank the reviewer for their suggestion. We agree the terminology might have been confusing and therefore changed the label of “action continuation” into the “continuation of behaviour” throughout our manuscript and in Figure 2B.

□ Although I quite like figures 1, I am wondering if the reference line at $p = .5$ is really necessary? There is no baseline probability of .5 and the line somewhat interferes with some of the datapoints (in particular with those of the chimpanzees).

Following both reviewers’ suggestions, the line has been removed and a revised version of the figure has been uploaded.

References related to response to reviews

- Bangerter, A., Chevalley, E., Derouwaux, S., 2010. Managing third-party interruptions in conversations: Effects of duration and conversational role. *J. Lang. Soc. Psychol.* 29, 235–244. <https://doi.org/10.1177/0261927X09359591>
- Chevalley, E., Bangerter, A., 2010. Suspending and reinstating joint activities with dialogue. *Discourse Process.* 47, 263–291. <https://doi.org/10.1080/01638530902959935>
- Clark, H.H., 2006. Social actions, social commitments, in: Enfield, N.J., Levinson, S.C. (Eds.), *Roots of Human Sociality: Culture, Cognition and Interaction*. Berg, Oxford, England, pp. 126–150.
- Clay, Z., de Waal, F.B.M., 2013. Bonobos respond to distress in others: consolation across the age spectrum. *PLoS One* 8, e55206.
- Gräfenhain, M., Behne, T., Carpenter, M., Tomasello, M., 2009. Young children’s understanding of joint commitments. *Dev. Psychol.* 45, 1430–1443. <https://doi.org/10.1037/a0016122>
- Gräfenhain, M., Carpenter, M., Tomasello, M., 2013. Three-year-olds’ understanding of the consequences of joint commitments. *PLoS One* 8, e73039. <https://doi.org/10.1371/journal.pone.0073039>
- Graham, K.E., Furuichi, T., Byrne, R.W., 2017. The gestural repertoire of the wild bonobo (*Pan paniscus*): a mutually understood communication system. *Anim. Cogn.* 20, 171–177. <https://doi.org/10.1007/s10071-016-1035-9>
- Hare, B., Melis, A.P., Woods, V., Hastings, S., Wrangham, R., 2007. Tolerance allows bonobos to outperform chimpanzees on a cooperative task. *Curr. Biol.* 17, 619–623. <https://doi.org/10.1016/j.cub.2007.02.040>
- Hare, B., Wobber, V., Wrangham, R., 2012. The self-domestication hypothesis: evolution of bonobo psychology is due to selection against aggression. *Anim. Behav.* 83, 573–585. <https://doi.org/10.1016/J.ANBEHAV.2011.12.007>
- Heesen, R., Bangerter, A., Zuberbühler, K., Iglesias, K., Neumann, C., Pajot, A., Perrenoud, L., Guéry, J.-P., Rossano, F., Genty, E., 2021. Assessing joint commitment as a process in great apes. *iScience* 102872. <https://doi.org/10.1016/J.ISCI.2021.102872>
- Heesen, R., Bangerter, A., Zuberbühler, K., Iglesias, K., Rossano, F., Guéry, J.P., Genty, E., 2020. Bonobos engage in joint commitment. *Sci. Adv.* 6, eabd1306. <https://doi.org/DOI:10.1126/sciadv.abd1306>
- Herrmann, E., Hare, B., Call, J., Tomasello, M., 2010. Differences in the cognitive skills of bonobos and chimpanzees. *PLoS One* 5, e12438. <https://doi.org/10.1371/journal.pone.0012438>
- Hobaiter, C., Byrne, R.W., 2011. Serial gesturing by wild chimpanzees: its nature and function for communication. *Anim. Cogn.* 14, 827–838. <https://doi.org/10.1007/s10071-011-0416-3>

- Hohmann, G., Fruth, B., 2008. New records on prey capture and meat eating by bonobos at Lui Kotale, Salonga National Park, Democratic Republic of Congo. *Folia Primatol.* 79, 103–110. <https://doi.org/10.1159/000110679>
- Kachel, U., Svetlova, M., Tomasello, M., 2019. Three- and 5-year-old children's understanding of how to dissolve a joint commitment. *J. Exp. Child Psychol.* 184, 34–47. <https://doi.org/10.1016/J.JECP.2019.03.008>
- Kano, F., Hirata, S., Call, J., 2015. Social attention in the two species of Pan: Bonobos make more eye contact than chimpanzees. *PLoS One* 10, e0129684. <https://doi.org/10.1371/journal.pone.0129684>
- Kret, M.E., Jaasma, L., Bionda, T., Wijnen, J.G., 2016. Bonobos (*Pan paniscus*) show an attentional bias toward conspecifics' emotions. *Proc. Natl. Acad. Sci. U. S. A.* 113, 3761–6. <https://doi.org/10.1073/pnas.1522060113>
- Liebal, K., Call, J., Tomasello, M., 2004. Use of gesture sequences in chimpanzees. *Am. J. Primatol.* 64, 377–396. <https://doi.org/10.1002/ajp.20087>
- Mayor, E., Bangerter, A., 2015. Managing perturbations during handover meetings: a joint activity framework. *Nurs. Open* 2, 130–140. <https://doi.org/10.1002/nop2.29>
- Rossano, F., 2013. Sequence organization and timing of bonobo mother-infant interactions. *Interact. Stud.* 14, 160–189. <https://doi.org/10.1075/is.14.2.02ros>
- Surbeck, M., Hohmann, G., 2008. Primate hunting by bonobos at LuiKotale, Salonga National Park. *Curr. Biol.* 18, R906–R907. <https://doi.org/10.1016/J.CUB.2008.08.040>
- Tan, J., Ariely, D., Hare, B., 2017. Bonobos respond prosocially toward members of other groups. *Sci. Rep.* 7, 14733. <https://doi.org/10.1038/s41598-017-15320-w>
- Tokuyama, N., Furuichi, T., 2016. Do friends help each other? Patterns of female coalition formation in wild bonobos at Wamba. *Anim. Behav.* 119, 27–35. <https://doi.org/10.1016/J.ANBEHAV.2016.06.021>
- Tokuyama, N., Sakamaki, T., Furuichi, T., 2019. Inter-group aggressive interaction patterns indicate male mate defense and female cooperation across bonobo groups at Wamba, Democratic Republic of the Congo. *Am. J. Phys. Anthropol.* 170, 535–550. <https://doi.org/10.1002/ajpa.23929>

Appendix D

Response to reviews

Editorial decision: Accepted, minor revision within 7 days (deadline extended)

Associate Editor Comments to Author (Dr Shinya Yamamoto):

I acknowledge that the authors have improved their manuscript considerably, and now it is close to acceptance for publication. However, the reviewer 1 still raised some important points. Please take his/her comments fully into consideration.

We are grateful to the editor and reviewers for their time in assessing our revisions and for the provisional acceptance of our revised manuscript. We have revised the manuscript in accordance with reviewer 1's remaining concerns, which we detail below. Overall, we hope these revisions have now resulted in our manuscript being fully acceptable for publication with RSOS. References used in our response are provided in a bibliography at the end of this letter.

Reviewer comments to Author:

Reviewer: 1

Comments to the Author(s)

I thank the authors for an excellent revision. I am happy to see care has been taken to respond to mine and the other reviewer's comments, and for the most part satisfied the concerns I had raised. I am especially glad to see that many of my more serious concerns were due to unclarity in the text rather than problems in the methods and analysis, which have now been satisfactorily described. This is an interesting paper and I think it represents a significant improvement from the first draft and contribution to the literature, though remain some important points I think should be raised before I can fully endorse publication.

Most significantly, while the authors now clarify based on my previous review they analyzed only external interruptions as opposed to natural ends, this term was not operationalized clearly. Without a definition for this, it is impossible to evaluate the rigour of these observations. While in the introduction the authors say "we recorded the apes' behaviours following both internal causes (partner(s) taking a break) and external causes (partner(s) stopping the activity due to a noise, conspecific interrupting, or other kindred events) of interruptions," "kindred events" is not specific enough to evaluate as a definition. In the methods you state "stop all movements characteristics of the activity, and produce at least one glance away from the activity" as part of the definition, but this would always be observed in a natural end as well (though I know this distinguishes between pause and interruption which I appreciate identifying). An operational definition of interruption compared to natural end is essential to be found in the methods.

We thank the reviewer for their comment and have now added a clearer definition of what distinguishes potential natural endings from interruptions. The definition was added to introduction (LL 98-105), method (LL 258-267) and Table 1 in our results.

I appreciate the very engaging and interesting discussion about triadic joint commitments in the discussion, though I think some mention of this should be included in the introduction as well when discussing what separates your study from others. While it is true that many previous studies may lack ecological validity, it is also a possibly important difference that they almost entirely focus on triadic tasks and thus may be unsuitable in that way (or at least leave an open question about whether great apes engage in dyadic joint commitment). I think some mention of this would help readers follow another key difference between yours and previous studies, as well as link more smoothly to the interesting discussion about it later on.

We added a few lines on this matter to our introduction (LL 45-46 and 59-65).

It is not necessary to include this in the manuscript, but have you tried analyzing the likelihood of resumption by social rank/social bonds based on timing? With the very short intervals, there may be much smaller likelihood of abandoning the activity in general, but as intervals become longer it will require more dedicated effort to resume a joint activity. Perhaps floor effects in the shortest intervals limit the rank/bond differences, but that greater differences would be more noticeable when looking at the longer intervals (or perhaps interruptions involving moving from the location and returning using your partner separation variable). In cases where it is noticeable, if it can be operationalized well, it may also be interesting to look at unilateral interruptions as compared to both partners getting distracted, similar to your earlier study involving experimental interruptions. There may not be sufficient data for this, but it will be an interesting question in the future. If the majority of cases came from distractions to both partners (which may be less salient than your earlier experimental interruptions which seem to have prompted suspension communication and high resumption) then there may be less importance of “politeness” or its equivalent since it is mutual, compared to one partner getting distracted and attempting to resume.

We thank the reviewer for these interesting ideas and we agree that this would be an interesting question to address in the future. However, in the current dataset we did not record whether one or both partners were responsible for interrupting the interaction following external causes. Thus, we are unable to test whether the likelihood of resumption depends on the duration of interruption, and how this might interact with the relationship between partners.

Thank you for including more details about DSI calculation. One question I have is how this calculation compensates for differing coresidency in the San Diego group you mention you performed? It is a bit confusing here as well and is an important point for social relations with differing grouping so more details here would be appreciated.

The DSI computations were performed using an R script published on Github (Neumann, 2016 <https://github.com/gobbios/socialindices>). The formula stated in the manuscript is a general computation of DSI, which follows only after having controlled of co-residency and observation time using functions of the R script itself. Hence, those parameters cannot directly be indicated in the mathematical expression of the formula. The link to the R package entails details about all the functions relevant to performing the DSI analysis, as well as a step-by-step manual. Since stating the details of the R functions goes beyond the scope of our method section, we have merely added the link to the script in the methods (LL 204-205).

Regarding the choice of minimum time to count as an interruption, I appreciate the greater discussion, though I wonder how relevant the signalling response time is as the basis to set this definition. Waiting for a response after a signal seems a very different thing than time at which an interruption is recognized as an interruption.

We agree with reviewer 1 that response waiting during a communicative exchange is not equivalent to action suspension (suspending the movements typical of the activity) during an social interaction. However, we believe that using a similar criteria (here 3 s) for a minimal time window is still relevant. “Response waiting” and “action suspension” are comparable in the sense that they both represent a noticeable suspension of behavior (e.g., a minimum time lapse between two signalling bouts, or between two actions of an ongoing interaction where actions are performed continuously like in play and grooming).

I would also recommend including another histogram with more bins, since the first bar in the histogram is far larger than any others, but you suggest only a small proportion of interruptions were <5s. A histogram focusing perhaps on these first bars with many more bins would clarify that a bit more and be more convincing.

A new histogram with smaller time bins has been added as Fig. S3. The small proportions of interruptions <5s is now visible.

In the paragraph starting on line 429 and elsewhere it would be good to mention again you mean specifically the same style of behaviour (location or play type) rather than just the general categories of behaviour (grooming, play). While you earlier emphasize this point of the analysis, for a naïve reader who does not focus on all the details it may be confusing. Play and grooming are themselves behaviours, so you should be clear you mean an even higher level of specificity.

Specification was added (LL 448-449).

Another point that I do not think is necessary to include in the MS is whether the fact that social behaviours can be rewarding is an important difference from your solitary control in previous studies. If some rewarding individual behaviour is interrupted, they may resume more, perhaps the fact that social behaviours are rewarding, combined with recency bias, rather than pure commitment can be an alternative understanding which needs some controlled conditions to rule out.

Since the discussion on this matter was added in response to a point raised by Reviewer 2, we would prefer to keep it in the manuscript. Social rewards are probably drivers of joint commitment in humans as well; we do not necessarily see why social rewards would rule out evidence for joint commitment. If recency biases had led to higher resumption rates, then this would have to be the case for both social and non-social activities, which was not the case in our previous study on bonobos (Heesen, Bangerter, Zuberbühler, Iglesias, Rossano, Guéry, and Genty, 2020).

The point about triadic joint commitment may fit into the final paragraph as well, since collaborative extractive foraging may depend more on this triadic joint commitment than dyadic. It is also worth noting that group hunting, female alliances attacking male bonobos, and chimpanzee border patrols may be more triadic than dyadic in some sense, even though the attention target is not always in direct sight (prey animals, troublesome males, or outgroup members, respectively). The dyadic commitments of bonobos and chimpanzees you study may be closely related to social bond development/maintenance, coalition formation, and group stability/cohesion, while triadic joint commitment warrants future study based on these behaviours observed in chimpanzees and bonobos.

We thank the reviewer for this suggestion, but prefer to keep this discussion about triadic and dyadic forms of commitment rather short; the reason is because, as previously mentioned, it is entirely speculative at this point whether there are different forms of joint commitment in dyadic versus triadic interactions. We believe that such a discussion goes beyond the scope of this paper (i.e., it was not our major focus or part of our predictions).

Another question is whether you ever saw play or grooming in groups? I often have seen chimpanzees play with more than one individual and regularly see bonobos in grooming parties with more than 2 individuals, were these ever observed in your groups? If so, how were they coded and analyzed?

Play and grooming interactions involving more than 2 individuals were also observed in the studied groups. However we only focused on dyadic interactions for our analyses in this study.

While both species hunt monkeys, and there is likely under reporting of the frequency in bonobos, I think it should be pointed out that there appears to be a species difference here. Chimpanzees have been observed hunting more frequently, and in one of the bonobo papers you cite (Hohmann and Fruth 2008) there is evidence for meat eating but no evidence for collaborative hunting

presented. There are several bonobo groups which have been observed daily for years but the frequency of observation of group hunting is far lower, never reported at some sites. The prey choice also differs, where bonobos tend to prefer individual hunting of duikers, chimpanzees prefer group hunting of monkeys (though these are of course far from exclusive tendencies). Bonobos have not been observed hunting red colobus monkeys, the preferred prey of chimpanzees, but instead have been observed mutual grooming with them. The species seem to have different kinds of social intelligence here, and their collaborative hunting should not be lumped together without at least a disclaimer that it is seen more rarely in bonobos.

To our knowledge, there is currently no direct, systematic comparison between the frequency of group hunting in bonobos and chimpanzees; and as pointed out by reviewer 1 the evidence for bonobo hunting is fairly recent and underrepresented in the literature. At least one of the reference cited (Surbeck and Hohmann, 2008) shows that bonobos hunt in parties and not alone, with coordinated movements between individuals. We thus gently disagree with the reviewer on the necessity to distinguish these two sets of evidence. More research and direct observations are needed to understand whether there are differences in the complexity of chimpanzee and bonobo hunting techniques.

A supplementary video of an example of short interruption would be beneficial, currently all videos show long interruptions, which although possibly the most interesting are not representative of all your observations, so some examples on the shorter end should be included.

We have now added a movie showing an interruption of 4.7 seconds (movie S8).

I thank the authors for their care in responding and an excellent revision.

Reviewer: 2

Comments to the Author(s)

I think that the authors did a good job clarifying the operationalizations and enriching the discussion, which now more clearly and explicitly communicates the contribution of the current work to the field.

I did spot a small typo (Line 436: "evinced" should be evidenced) that will need to get fixed at some point during the process.

Done.

References

- Genty, E., Heesen, R., Rossano, F., Zuberbühler, K., Guery, J., Bangerter, A., 2020. How apes get into and out of joint actions: precursors of shared intentionality? *Interact. Stud.* 21, 353–386. <https://doi.org/https://doi.org/10.1075/is.18048.gen>
- Heesen, R., Bangerter, A., Zuberbühler, K., Iglesias, K., Rossano, F., Guéry, J.P., Genty, E., 2020. Bonobos engage in joint commitment. *Sci. Adv.* 6, eabd1306. [https://doi.org/DOI: 10.1126/sciadv.abd1306](https://doi.org/DOI:10.1126/sciadv.abd1306)
- Mercier, S., Neumann, C., van de Waal, E., Chollet, E., Meric de Bellefon, J., Zuberbühler, K., 2017. Vervet monkeys greet adult males during high-risk situations. *Anim. Behav.* 132, 229–245. <https://doi.org/10.1016/J.ANBEHAV.2017.07.021>
- Neumann, C., 2016. Socialindices: Social and association indices. Unpublished R package (v.0.46-7).
- Surbeck, M., Hohmann, G., 2008. Primate hunting by bonobos at LuiKotale, Salonga National Park. *Curr. Biol.* 18, R906–R907. <https://doi.org/10.1016/J.CUB.2008.08.040>